# FisB relies on homo-oligomerization and lipid binding to catalyze membrane fission in bacteria

Ane Landajuela[1,2☯*], Martha Braun[2,3☯], Christopher D. A. Rodrigues[4], Alejandro Martínez-Calvo[5], Thierry Doan[6], Florian Horenkamp[7¤a], Anna Andronicos[1¤b], Vladimir Shteyn[1,2], Nathan D. Williams[2,7], Chenxiang Lin[2,7], Ned S. Wingreen[8,9], David Z. Rudner[10], Erdem Karatekin[1,2,3,11*]

1 Cellular and Molecular Physiology, Yale University, New Haven, Connecticut, United States of America, 2 Nanobiology Institute, Yale University, West Haven, Connecticut, United States of America, 3 Molecular Biophysics and Biochemistry, Yale University, New Haven, Connecticut, United States of America, 4 iThree Institute, University of Technology Sydney (UTS), Australia, 5 Grupo de Mecánica de Fluidos, Universidad Carlos III de Madrid, Spain, 6 Laboratoire d'Ingénierie des Systèmes Macromoléculaires, Aix-Marseille Université, Marseilles, France, 7 Cell Biology, Yale University, New Haven, Connecticut, United States of America, 8 Department of Molecular Biology, Princeton University, Princeton, New Jersey, United States of America, 9 Lewis-Sigler Institute for Integrative Genomics, Princeton University, Princeton, New Jersey, United States of America, 10 Department of Microbiology, Harvard Medical School, Boston, Massachusetts, United States of America, 11 Université de Paris, SPPIN—Saints-Pères Paris Institute for the Neurosciences, Centre National de la Recherche Scientifique (CNRS), Paris, France

☯ These authors contributed equally to this work.
¤a Current address: Pfizer, New London, Connecticut, United States of America
¤b Current address: Department of Biological Chemistry, School of Medicine, University of California, Irvine, California, United States of America
* ane.landajuela@yale.edu (AL); erdem.karatekin@yale.edu (EK)

**Data Availability Statement:** All relevant data are within the paper and its Supporting Information files.

## Abstract

Little is known about mechanisms of membrane fission in bacteria despite their requirement for cytokinesis. The only known dedicated membrane fission machinery in bacteria, fission protein B (FisB), is expressed during sporulation in *Bacillus subtilis* and is required to release the developing spore into the mother cell cytoplasm. Here, we characterized the requirements for FisB-mediated membrane fission. FisB forms mobile clusters of approximately 12 molecules that give way to an immobile cluster at the engulfment pole containing approximately 40 proteins at the time of membrane fission. Analysis of FisB mutants revealed that binding to acidic lipids and homo-oligomerization are both critical for targeting FisB to the engulfment pole and membrane fission. Experiments using artificial membranes and filamentous cells suggest that FisB does not have an intrinsic ability to sense or induce membrane curvature but can bridge membranes. Finally, modeling suggests that homo-oligomerization and *trans*-interactions with membranes are sufficient to explain FisB accumulation at the membrane neck that connects the engulfment membrane to the rest of the mother cell membrane during late stages of engulfment. Together, our results show that FisB is a robust and unusual membrane fission protein that relies on homo-oligomerization, lipid binding, and the unique membrane topology generated during engulfment for localization and membrane scission, but surprisingly, not on lipid microdomains, negative-curvature lipids, or curvature sensing.

**Funding:** This work was supported by the National Institute of General Medical Sciences, R01GM114513 (EK); theNational Institute of Neurological Disorders and Stroke, R01NS113236 (EK); the National Institutes of Health (US), DP2GM114830 (CL); the National Institute of General Medical Sciences, R01GM132114 (CL); the Office of Extramural Research, National Institutes of Health, T32-EB09941 (NDW); Yale University Predoctoral Fellowship (MB); the National Science Foundation, Center for the Physics of Biological Function (PHY-1734030) (NSW) and the Spanish Ministry of Education, Culture and Sport, Ayudas para la Formación de Profesorado Universitario, Grant No. FPU16/0256 (AM-C). The funders had no role in study design, data collection and analysis, decision to publish, or preparation of the manuscript.

**Competing interests:** The authors have declared that no competing interests exist.

**Abbreviations:** CCCP, carbonyl cyanide m-chlorophenyl hydrazone; CFU, colony-forming unit; CL, cardiolipin; cryo-EM, cryogenic electron microscopy; DMC, dim mobile cluster; DOPC, 1,2-dioleoyl-sn-glycero-3-phosphocholine; DOPE, 1,2-dioleoyl-sn-glycero-3-phosphoethanolamine; DOPS, 1,2-dioleoyl-sn-glycero-3-phospho-L-serine; DSM, DS medium; ECD, extracytoplasmic domain; eggPC, egg L-α-phosphatidylcholine; EM, electron microscopy; ESCRT-III, endosomal sorting complex required for transport III; FisB, fission protein B; FMM, functional membrane microdomain; GUV, giant unilamellar vesicle; iFluor555-FisB ECD, iFluor555-labeled FisB ECD; ISEP, intense spot at the engulfment pole; ITO, indium tin oxide; LAS X, Leica Application Suite X; LJ, Lennard–Jones; LoG, Laplacian of Gaussian; mGFP, monomeric EGFP; MSD, mean-squared displacement; mYFP, monomeric YFP; PC, phosphatidylcholine; PE, phosphatidylethanolamine; PG, phosphatidylglycerol; PS, phosphatidylserine; PTFE, Polytetrafluoroethylene; RBS, ribosome binding site; SCAM, substituted cysteine accessibility method; SEC, size exclusion chromatography; SNARE, soluble N-ethylmaleimide-sensitive-factor attachment protein receptor; SUV, small unilamellar vesicle; TCA, trichloroacetic acid; TLC, thin-layer chromatography; TMD, transmembrane domain.

## Introduction

Membrane fission is a fundamental process required for endocytosis [1], membrane trafficking [2], enveloped virus budding [3], phagocytosis [4], cell division [5], and sporulation [6–8]. During membrane fission, an initially continuous membrane divides into 2 separate ones. This process requires dynamic localization of specialized proteins, which generate the work required to merge membranes [9–13]. Dynamin [14] and the endosomal sorting complex required for transport III (ESCRT-III) catalyze many eukaryotic membrane fission reactions [15]. Both fission machineries bind acidic lipids, assemble into oligomers, and use hydrolysis of a nucleoside triphosphate (GTP or ATP) to achieve membrane fission. However, membrane fission can also be achieved by friction [16], stress accumulated at a boundary between lipid domains [17], forces generated by the actomyosin network [18–21], or protein crowding [22]. By contrast, very little is known about membrane fission in bacteria, even though they rely on membrane fission for every division cycle.

We previously found that fission protein B (FisB) is required for the final membrane fission event during sporulation in *Bacillus subtilis* [23]. When nutrients are scarce, bacteria in the orders Bacillales and Clostridiales initiate a developmental program that results in the production of highly resistant endospores [24]. Sporulation starts with an asymmetric cell division that generates a larger mother cell and a smaller forespore (Fig 1A). The mother cell membranes then engulf the forespore in a process similar to phagocytosis. At the end of engulfment, the leading membrane edge forms a small pore. Fission of this membrane neck connecting the engulfment membrane to the rest of the mother cell membrane releases the forespore, now surrounded by 2 membranes, into the mother cell cytoplasm (Fig 1A and 1B). At this late stage, the mother nurtures the forespore as it prepares for dormancy. Once mature, the mother cell releases the spore into the environment through lysis. Spores can withstand heat, radiation, drought, antibiotics, and other environmental assaults for many years [25–28]. Under favorable conditions, the spore will germinate and restart the vegetative life cycle.

Conserved among endospore-forming bacteria, FisB is a mother cell transmembrane protein expressed under the control of the transcription factor, $\sigma^E$, after asymmetric division [29]. In sporulating cells lacking FisB, engulfment proceeds normally, but the final membrane fission event, detected using a lipophilic dye, is impaired [23] (Fig 1C and 1F, S1 Appendix Fig A, panel A). During engulfment, FisB fused to a fluorescent protein forms dim mobile clusters (DMCs) in the engulfment membrane (Fig 1D and 1E, S1 Movie). When the engulfing membranes reach the cell pole, approximately 3 hours (t = 3 hours) after the onset of sporulation, a cluster of FisB molecules accumulates at the pole forming a more intense, immobile focus, where and when fission occurs (Fig 1D and 1E, S2 Movie).

We had previously reported [23] that FisB interacts with cardiolipin (CL), a lipid enriched at cell poles [30–32] whose levels increase during sporulation [33] and is implicated in membrane fusion [34–36] and fission reactions [37]. In addition, CL was reported to act as a landmark for the polar recruitment of the proline transporter ProP and the mechanosensitive channel MscSm [38,39]. Thus, it seemed plausible that CL could act as a landmark to recruit FisB to the membrane fission site and facilitate membrane fission. Apart from this hypothesis, no information has been available about how FisB localizes to the membrane fission site and how it may drive membrane scission.

Here, we determined the requirements for FisB's subcellular localization and membrane fission during sporulation. Using quantitative analysis, we find small clusters of approximately 12 FisB molecules diffuse around the mother cell membrane and approximately 40 copies of FisB accumulate at the fission site as an immobile cluster to mediate membrane fission. When FisB expression was lowered, approximately 6 copies of FisB were sufficient to drive

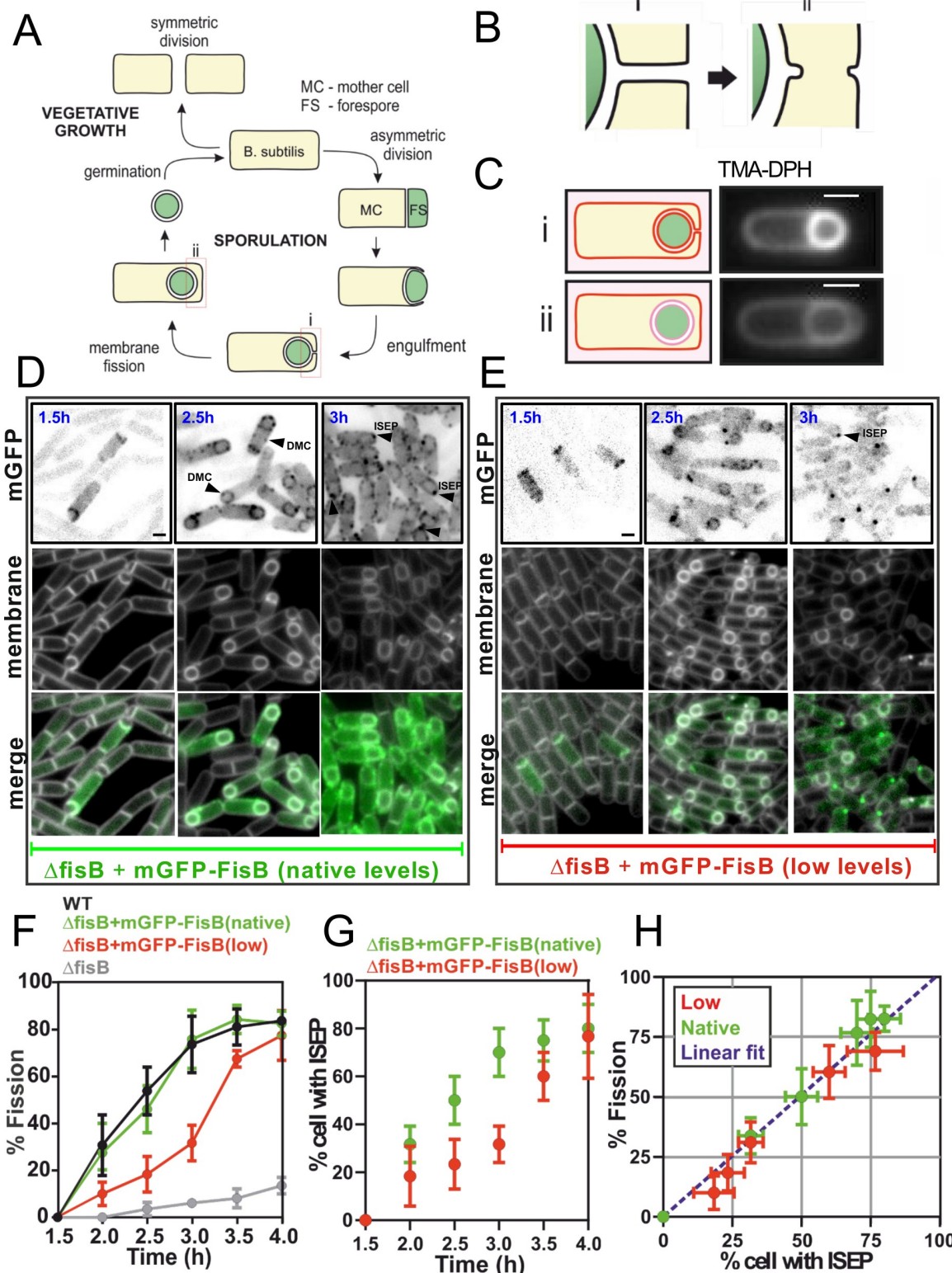

**Fig 1. Membrane fission during sporulation is nearly always accompanied by accumulation of a FisB cluster at the fission site. (A)** Vegetatively growing cells enter sporulation when nutrients become scarce. Asymmetric division creates an FS and an MC. The MC engulfs the FC in a phagocytosis-like event. At the end of engulfment, a membrane neck connects the engulfment membrane to the rest of the MC (i). Fission of the neck (ii) releases the FS, now surrounded by 2 membranes, into the MC cytoplasm. Once the FS becomes a

mature spore, the MC lyses to release it. **(B)** The membrane fission step shown in more detail. **(C)** Detection of membrane fission. The lipophilic dye TMA-DPH does not fluoresce in the aqueous solution and crosses membranes poorly. If membrane fission has not yet taken place, the dye has access to the engulfment, FS, and MC membranes thus shows intense labeling where these membranes are adjacent to one another (i). If fission has already taken place, the dye labels internal membranes poorly (ii). **(D)** Images show mGFP-FisB (strain BAM003, native expression level) at indicated times during sporulation. Membranes were visualized with TMA-DPH. Examples of sporulating cells with mGFP-FisB enriched at the septum (1.5 hours), forming DMC (2 hours) and with a discrete mGFP-FisB focus at the cell pole (intense spot at engulfment pole, ISEP, 3 hours), are highlighted with arrowheads. **(E)** Similar to D, but using a strain (BAL003) that expresses mGFP-FisB at lower levels in a ΔfisB background. **(F)** Time course of membrane fission for WT cells, ΔfisB cells, or ΔfisB cells complemented with mGFP-FisB expressed at native (BAM003) or low levels (BAL003). Lower expression of mGFP-FisB leads to a delay in membrane fission kinetics. **(G)** The percentage of cells with an ISEP for low and native level expression of mGFP-FisB as a function of time into sporulation. **(H)** Correlation between percentage of cells that have undergone fission and percentage of cells having an ISEP for all time points shown in F and G. The fitted dashed line passing through the origin has slope 1.06 ($R^2 = 0.9$). Scale bars represent 1 μm. DMC, dim mobile cluster; FisB, fission protein B; FS, forespore; ISEP, intense spot at the engulfment pole; MC, mother cell; mGFP, monomeric EGFP; WT, wild-type.

membrane fission, but fission took longer. Unexpectedly, FisB dynamics and membrane fission are independent of both CL and phosphatidylethanolamine (PE), another lipid implicated in membrane fusion and fission. We found that FisB binds phosphatidylglycerol (PG) with comparable affinity as CL, after adjusting for charge density. Thus, we suspect that, as a more abundant lipid in the cell, PG can substitute for CL to bind FisB. We tested other factors that may be important for the subcellular localization of FisB and membrane fission. We found that FisB dynamics are independent of flotillins, which organize bacterial membranes into functional membrane microdomains (FMMs) [40], cell wall synthesis machinery, and proton or voltage gradients across the membrane. Using mutagenesis, we show that both FisB oligomerization and binding to acidic lipids are required for proper localization and membrane fission. *B. subtilis ΔfisB* cells were partially complemented by *Clostridium perfringens* FisB, despite only approximately 23% identity between the 2 proteins, suggesting a common localization and membrane fission mechanism based on a few conserved biophysical properties. The membrane neck that eventually undergoes fission and where FisB accumulates is the most highly curved membrane region in the late stages of engulfment. Thus, FisB could potentially localize at the membrane neck due to a preference for highly curved membrane regions. We tested this possibility in experiments with both artificial giant unilamellar vesicles (GUVs) and live cells. Surprisingly, these experiments failed to reveal any intrinsic affinity of FisB for highly curved membranes. However, we found that FisB bridges membranes and accumulates at membrane adhesion sites. Using modeling, we found that self-oligomerization of FisB, coupled with its ability to bridge negatively charged membranes, is sufficient to explain its localization to the membrane neck. Thus, proteins can localize to highly curved membrane regions through mechanisms independent of intrinsic curvature sensitivity. Together, these results suggest that FisB–FisB and FisB–lipid interactions, combined with the unique membrane topology generated at the engulfment pole during sporulation, provide a simple mechanism to recruit FisB to mediate membrane fission independent of other factors.

## Results

### Membrane fission occurs in the presence of a cluster of FisB molecules

To correlate FisB dynamics with membrane fission, we devised a labeling strategy that allowed us to monitor both simultaneously, using a modified version of a fission assay developed previously [41]. In this assay, synchronous sporulation is induced by placing *B. subtilis* cells in a nutrient-poor medium. At different time points after the nutrient downshift, aliquots are harvested from the culture, stained with the lipophilic membrane dye FM4-64, mounted on an agar pad, and imaged using fluorescence microscopy. The dye is virtually nonfluorescent in

the medium, and it cannot cross the cell membrane. Thus, before fission, FM4-64 labels the outer leaflet of both the mother cell and the forespore membranes. After fission, only the outer leaflet of the mother cell is labeled (S1 Appendix Fig A, panel B). Because post-fission cells and cells that never entered sporulation are labeled identically, in addition to FM4-64, a fluorescent protein is expressed in the forespore under the control of the forespore-specific transcription factor $\sigma^F$ to distinguish between the 2 cell types [42] (S1 Appendix Fig A, panel B). This makes it challenging to monitor FisB dynamics simultaneously, which requires a third channel. As an alternative, we used another lipophilic dye, TMA-DPH, that has partial access to internal membranes but can distinguish between pre- and post-fission stages without the need for a forespore reporter [23] (Fig 1C, S1 Appendix Fig A, panels D–G). Using TMA-DPH as the fission reporter, we quantified the percentage of cells that have undergone fission as a function of time, for wild-type, *fisB* knock-out (*ΔfisB*, strain BDR1083, see S1 Appendix Table B for strains used), and *ΔfisB* cells complemented with FisB fused to monomeric EGFP (mGFP-FisB, strain BAM003) as shown in Fig 1D and 1F. These kinetic measurements reproduced results obtained using FM4-64 (S1 Appendix Fig A, panel C). Thus, TMA-DPH can be used as a faithful reporter of membrane fission, leaving a second channel for monitoring dynamics of FisB fused to a fluorescent reporter.

In the experiments of Fig 1D and 1F, we simultaneously monitored dynamics of mGFP-FisB and membrane fission. We found that membrane fission is almost always accompanied by an intense, immobile mGFP-FisB signal at the engulfment pole (Fig 1D, time = 3 hours into sporulation). This intense spot at the engulfment pole (ISEP) is distinct from the DMCs that appear at earlier times elsewhere (Fig 1D). By 3 hours into sporulation, around 70% of the cells expressing mGFP-FisB at native levels had an ISEP (Fig 1G), a number that was close to the percentage of cells that had undergone fission by then (Fig 1F). Scoring individual cells, we found that >90% (212/235) of cells that had undergone membrane fission also had an ISEP.

We also monitored membrane fission and mGFP-FisB signals in a strain with lower FisB expression. Here, lower FisB expression is achieved by reducing the spacing between the ribosome binding site (RBS) and the adenine-thymine-guanine (ATG) start codon [43]. In this strain (BAL003), there was an initial delay in the fraction of cells that had undergone fission, but fission accelerated after t = 3 hours to reach near wild-type levels at around t = 4 hours (Fig 1E and 1F). The fraction of cells with an ISEP followed a similar pattern (Fig 1G). The fraction of cells that had undergone fission at a given time was strongly correlated with the fraction of cells with an ISEP at that time (Fig 1H). Scoring individual cells, we found that >93% (258/277) of cells that had undergone membrane fission had an ISEP. We conclude that membrane fission occurs in the presence of a large immobile cluster of FisB molecules at the site of fission.

## About 40 FisB molecules accumulate at the engulfment pole to mediate membrane fission

We asked how many copies of FisB are recruited to the engulfment pole at the time of membrane fission and how this number is affected by the expression level. For this quantification, we used DNA origami-based fluorescence standards we recently developed [44]. These standards consist of DNA rods (approximately 410-nm long and 7-nm wide) labeled with AF647 at both ends and a controlled number of mEGFP molecules along the rod (Fig 2A).

DNA origami standards carrying different mEGFP copies were imaged using wide-field fluorescence microscopy (Fig 2B). For each type of rod, the average total fluorescence intensity of single rods was computed and plotted against the number of mEGFP molecules per rod, generating the calibration curve in Fig 2D. We generated *B. subtilis* cells expressing

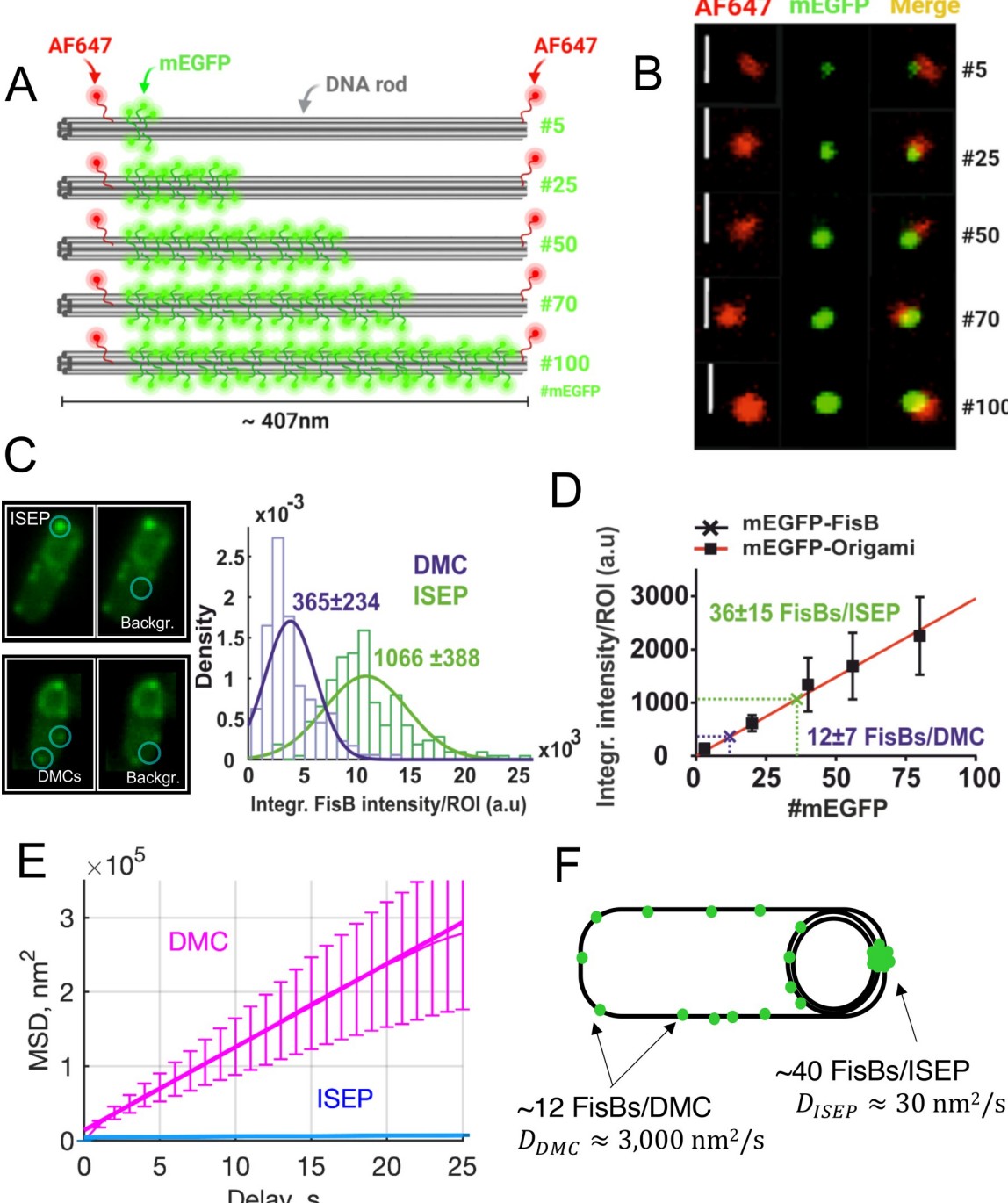

**Fig 2. Estimation of mEGFP-FisB copies at the engulfment pole at t = 3 hours using DNA origami calibration standards and mobility of FisB clusters.** (**A**) Simplified schematic of the DNA origami-based mEGFP standards used in this study. Using DNA origami, DNA rods bearing AF647 at both ends and the indicated numbers of mEGFP molecules along the rod were designed. In the actual rods, the labeling efficiency was found to be approximately 80%, so the actual copies of mEGFP per rod were 4, 20, 40, 56, and 80. (**B**) Representative wide-field images of the DNA origami-based mEGFP standards used in this study. Bars are 1 μm. (**C**) Distributions of total fluorescence intensities (sum of pixel values) for the ISEP and the DMC. Background was defined individually for every cell where an ISEP or DMC intensity measurement was performed. Examples are shown on the left. (**D**) Total fluorescence intensity (sum of pixel values) for DNA origami rods as a function of mEGFP copy numbers. The best fit line passing through the origin has slope 29.56 au/mEGFP ($R^2$ = 0.97). The total intensity of the ISEP and DMCs correspond to approximately 40 and approximately 12 copies of mEGFP, respectively. (**E**) MSD as a function of delay time for DMCs (magenta) and ISEPs (blue). Cells expressing mGFP-FisB (strain BAM003) were imaged using time-lapse microscopy. A total of 45 cells from 10 different movies at t = 2.5 hours and 30 cells from 10

different movies at t = 3 hours after nutrient downshift were analyzed. (See S1 Movie for a representative single bacterium at t = 2.5 hours showing several mobile DMCs and S2 Movie for a representative single bacterium at t = 3 hours showing an immobile ISEP.) Fits to the initial 25 seconds (approximately 10% of delays) yielded $D_{DMC}$ = 2.80±0.05×10$^3$ nm$^2$/s (± 95% confidence interval, R$^2$ = 0.999, 24 tracks) and $D_{ISEP}$ = 2.80±0.51×10 nm^2/s (± 95% confidence interval, R$^2$ = 0.850, 25 tracks). **(F)** Summary of FisB copy number and cluster mobility estimation. DMC, dim mobile cluster; FisB, fission protein B; ISEP, intense spot at the engulfment pole; MSD, mean-squared displacement.

mEGFP-FisB at native levels (BAL001) in a *ΔfisB* background so that images of these cells obtained under identical imaging conditions as for the calibration curve in Fig 2D could be used to compute mEGFP-FisB copy numbers. We imaged mEGFP-FisB cells at t = 3 hours after sporulation was induced. From these same images, we estimated the total fluorescence of DMC and ISEP in *B. subtilis* cells as a sum of background-corrected pixel values (Fig 2C). Using the average values of these total intensities, we estimate approximately 40 copies at the ISEP and approximately 12 per DMC from the calibration in Fig 2D. From the total intensity of cells (S1 Appendix Fig B, panel E), we also estimate that there are approximately 1,000 FisB molecules per cell. Two independent estimates, based on *B. subtilis* calibration strains [45] and quantitative immunoblotting, resulted in slightly larger and smaller estimates of these copy numbers, respectively (S1 Appendix, S1 Appendix Figs B and C).

We tracked the DMC to estimate how rapidly they move. From the tracks, we calculated the mean-squared displacement (MSD) as a function of time (Fig 2E). The short-time diffusion coefficient estimated from the MSD is $D_{DMC}$≈2.8×10$^3$ nm$^2$/s (95% confidence interval CI = 2.76–2.85×10$^3$ nm$^2$/s). This value is comparable to the diffusivity of FloA and FloT clusters of approximately 100 nm with $D$≈6.9×10$^3$ and 4.1×10$^3$ μm$^2$/s, respectively [46]. By comparison, ISEP have $D_{ISEP}$≈28 nm$^2$/s (CI = 22.9–33.1 nm$^2$/s), 2 orders of magnitude smaller.

We performed similar estimations of FisB copy numbers for the low-expression strain (BAL004) (S1 Appendix Fig D). We found approximately 160 ± 66, 122 ± 51, or 83 ± 6 (±SD) copies per cell using *B. subtilis* standards, DNA origami, or the quantitative Western blotting methods, respectively. For the ISEP, we found 8 ± 2, 6 ± 2, or 5 ± 3 (±SD) copies of mGFP-FisB using the 3 approaches, respectively (S1 Appendix Table A). About 6% of the total mGFP-FisB signal accumulated in ISEP, close to the approximately 4% in the native-expression strain (S1 Appendix Fig D, panel E). The native expression were too dim to quantify reliably. Assuming DMCs to be approximately 3-fold dimmer than ISEP like in the native-expression strain, each DMC would contain 2 to 3 mGFP-FisB, just below our detection limit. Interestingly, lowering the total expression of FisB per cell approximately 8-fold resulted in an approximately 6-fold reduction in the average number of FisB molecules found at the membrane fission site. Thus, only approximately 6 copies of FisB are sufficient to mediate membrane fission, but only after some delay (Fig 1E and 1F).

In summary, approximately 40 FisB molecules accumulate at the fission site to mediate membrane fission. Only 3 to 4 DMCs need to reach the fission site to provide the necessary numbers. When FisB expression is lowered approximately 8-fold, approximately 6 FisB molecules accumulate at the engulfment pole to mediate membrane fission, but fission takes longer.

## FisB localization and membrane fission are independent of cardiolipin, phosphatidylethanolamine, and flotillins

To investigate how FisB is recruited to the membrane fission site, we began by testing a potential role for the cell wall remodeling machinery, the proton motive force, and the membrane potential and found none influenced FisB dynamics (S1 Appendix Results, S1 Appendix Fig F).

We then tested whether lipid microdomains play a role in recruitment to the site of fission. Previously, we reported that the recombinant, purified extracytoplasmic domain (ECD; see Fig 4A) of FisB interacts with artificial lipid bilayers containing CL [23]. To test if FisB–CL interactions could be important for the subcellular localization of FisB and membrane fission, we generated a strain (BAM234) that carries deletions of the 3 known CL synthase genes *ywnE (clsA)*, *ywjE (clsB)*, and *ywiE (clsC)* [47] (Fig 3A). The CL synthase–deficient strain did not contain detectable levels of CL at t = 3 hours after sporulation was initiated (Fig 3B). CL-deficient cells grew normally but had a reduction in sporulation efficiency as assayed by heat-resistant (20 minutes at 80˚C) colony forming units (CFUs; S1 Appendix Table B, S1 Appendix Fig E) [33]. A reduction in sporulation efficiency measured in this manner can be due to a defect at one or several steps during sporulation or germination. Importantly, the membrane fission time course of *ΔclsABC* cells was indistinguishable from those of wild-type cells (Fig 3C and 3D), indicating the defect in sporulation occurs at a stage after membrane fission. In addition, monomeric YFP (mYFP)-FisB localization and dynamics were similar in *ΔclsABC* (BAL037) and wild-type (BAL002) cells (Fig 3F–3H). The fraction of cells that had an ISEP, and the intensity of the ISEP, reflecting the number of FisB molecules recruited to the membrane fission site, were indistinguishable for wild-type and *ΔclsABC* cells (Fig 3G and 3H). We conclude that CL is not required for the subcellular localization of FisB or membrane fission.

Next, we tested a potential role for PE, another lipid implicated in membrane fusion and fission [50,51] and that forms microdomains [52]. We deleted the *pssA* gene, which encodes phosphatidylserine (PS) synthase that mediates the first step in PE synthesis (Fig 3A) to generate cells lacking PE (strain BAL031, Fig 3B). Kinetics of membrane fission during sporulation were identical in *ΔpssA* and wild-type cells (Fig 3D), indicating that PE does not play a significant role in membrane fission.

PE and CL domains in *B. subtilis* membranes tend to occur in the same subcellular regions [52], raising the possibility that CL and PE may compensate for each other. To test whether removing both CL and PE affects fission, we generated a quadruple mutant (BAL030) lacking both CL and PE (Fig 3B), leaving PG as the major phospholipid component of the membrane. Surprisingly, the quadruple mutant underwent fission with indistinguishable kinetics compared to wild type (Fig 3C and 3D). Thus, 2 lipids with negative spontaneous curvature and implicated in membrane fusion and fission reactions in diverse contexts have no significant role in membrane fission mediated by FisB during sporulation.

In addition to CL and PE microdomains, bacteria also organize many signal transduction cascades and protein–protein interactions into FMMs, loose analogs of lipid rafts found in eukaryotic cells [40]. The FMMs of *B. subtilis* are enriched in polyisoprenoid lipids and contain flotillin-like proteins, FloT and FloA, that form mobile foci in the plasma membrane [53,54]. FloT-deficient cells have a sporulation defect, but which sporulation stage is impaired is not known [46]. We observed that in *ΔfloA* (BAL035), but not *ΔfloT* (BAL036), cells are impaired in sporulation as assayed by heat-resistant CFUs (S1 Appendix Table B, S1 Appendix Fig E). However, when we monitored engulfment and membrane fission, we found both proceeded normally in *ΔfloA* cells (Fig 3D). Thus, the sporulation defect in *ΔfloA* cells lies downstream of engulfment and membrane fission. This was confirmed by blocking formation of FMMs during sporulation by addition of 50 µM zaragozic acid [55] to the sporulation medium which had no effect on the localization of mGPF-FisB (Fig 3E).

Together, these results imply that FisB-mediated membrane fission that marks the end of engulfment during sporulation is insensitive to the negative-curvature lipids CL and PE and to FloA/T-dependent lipid domains.

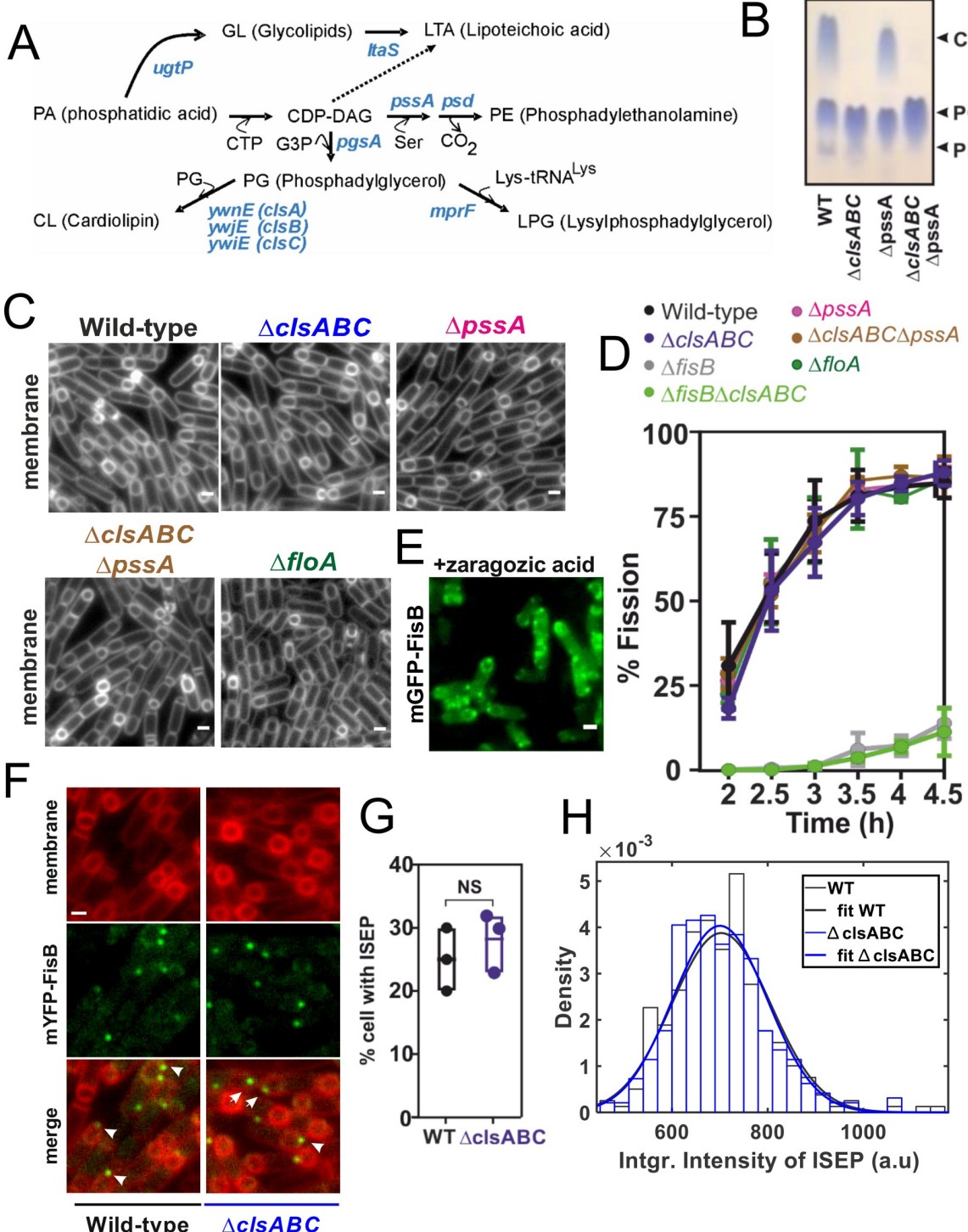

**Fig 3. Membrane fission is insensitive to membrane lipid composition.** (**A**) Pathways for membrane lipid synthesis in *B. subtilis*. Lipid synthetases responsible for each step are highlighted in blue. (**B**) TLC of the total lipid extracts of WT and indicated lipid synthesis–deficient cells. Cells were collected 3 hours after induction of sporulation by nutrient downshift. PL spots were visualized by staining with Molybdenum Blue spray reagent. Purified CL, PG, and PE were used as standards to identify the PLs of *B. subtilis*. Arrows indicate locations to which individual standards migrate. (**C**) Membranes from cells of the indicated genetic backgrounds were visualized with TMA-DPH at t = 3 hours. The images are from cells mounted on agarose pads containing sporulation medium. Bars are 1 μm. (**D**) Percentage of cells from indicated strains that have undergone membrane fission as a function of time after initiation of sporulation. For every strain, 150–220 cells from 3 independent experiments were analyzed at the indicated times during sporulation.

(E) mGFP-FisB (strain BAM003) treated with the squalene synthase inhibitor zaragozic acid, imaged at t = 3 hours. (F) Cells expressing mYFP-FisB (low expression levels) in either WT (BAL002) or in a CL-deficient strain (BAL037) at t = 3 hours. Membranes were visualized with the fluorescent dye TMA-DPH. Examples of sporulating cells with a discrete mYFP-FisB focus at the cell pole (ISEP) are highlighted (white arrows). Foci were semiautomatically selected with SpeckletrackerJ [48]. (G) The percentage of cells with an intense spot at engulfment pole for WT (BAL002) or CL-deficient (BAL037) mYFP-FisB expressing cells at t = 3 hours (low expression). For each strain, 150–220 cells from 3 independent experiments were analyzed. (H) Distributions of total fluorescence intensities (sum of pixel values) at ISEP for WT (BAL002) or CL-deficient (BAL037) mYFP-FisB cells at 3 hours into sporulation. For every strain, 150 ISEPs were analyzed. Scale bars are 1 μm. CL, cardiolipin; FisB, fission protein B; ISEP, intense spot at the engulfment pole; mYFP, monomeric YFP; PE, phosphatidylethanolamine; PG, phosphatidylglycerol; PL, phospholipid; TLC, thin-layer chromatography; WT, wild-type.

## FisB binds to acidic lipids

PG can substitute for CL as a binding partner for many proteins [56,57]. To see if this might also be the case for the FisB ECD, we quantified the affinity of this domain for both lipids.

Most, but not all, algorithms (S1 Appendix Fig G) predict FisB to possess a single transmembrane domain (TMD) with a small N-terminal cytoplasmic domain and a larger (23 kDa) ECD, as depicted in Fig 4A. We first confirmed this predicted topology using a cysteine accessibility assay [58] (S1 Appendix Fig H, Materials and methods, S1 Appendix Results). Our attempts to determine the structure of recombinant, purified FisB ECD were unsuccessful, but a computational model of FisB for residues 44 to 225, covering most of the ECD, is available [49] and is shown in Fig 4B. The model predicts a curved ECD structure, with approximately 3 nm and approximately 5 nm for the inner and outer radii of curvatures. The overall topology of FisB, with the predicted ECD structure, is depicted in Fig 4B.

We probed interactions of FisB ECD with PG using a liposome co-flotation assay, illustrated in Fig 4C. Purified recombinant, soluble FisB ECD (Fig 4A, bottom) was incubated with liposomes and subsequently layered at the bottom of a discontinuous density gradient. Upon equilibrium ultracentrifugation, the lighter liposomes float up to the interface between the 2 lowest density layers together with bound protein, while unbound protein remains at the bottom of the gradient. We collected fractions and determined the percentage of protein co-floated with liposomes using SDS-PAGE and densitometry, as shown in Fig 4D. We first determined that binding of FisB ECD to liposomes containing CL was not dependent on pH or the divalent ion $Ca^{2+}$ (S1 Appendix Fig I, panels F and G). By contrast, the fraction of liposome-bound protein decreased rapidly as the ionic strength increased (S1 Appendix Fig I, panel H). These results indicated binding was mainly electrostatic in nature.

At neutral pH, CL carries 2 negative charges, whereas PG and PS, a lipid not normally found in *B. subtilis* [59], carry only a single negative charge. If binding is mediated mainly by electrostatic interactions, then liposomes carrying PG or PS at 2 times the mole fraction of CL should bind nearly the same amount of FisB ECD, since the surface charge density would be the same. Indeed, similar amounts of FisB ECD were bound to liposomes carrying 30% CL, 60% PG, or 60% PS (Fig 4E). FisB ECD did not bind neutral phosphatidylcholine (PC) liposomes [23].

To quantify the affinity of recombinant soluble FisB ECD for CL versus PG, we then titrated liposomes containing 45 mole % CL or PG and measured binding of 100 nM FisB ECD (Fig 4F). In these experiments, we used iFluor555-labeled FisB ECD (iFluor555-FisB ECD) and detected liposome-bound protein using fluorescence rather than densitometry of SYPRO-stained gels, which extended sensitivity to much lower protein concentrations. The titration data were fit to a model to estimate the apparent dissociation constant, $K_d$ (see Materials and methods), which was 1.0 μM for CL (95% confidence interval CI = 0.7 to 2.1 μM) and 3.6 μM for PG, respectively (CI = 2.8 to 5.0, Fig 4F and 4G).

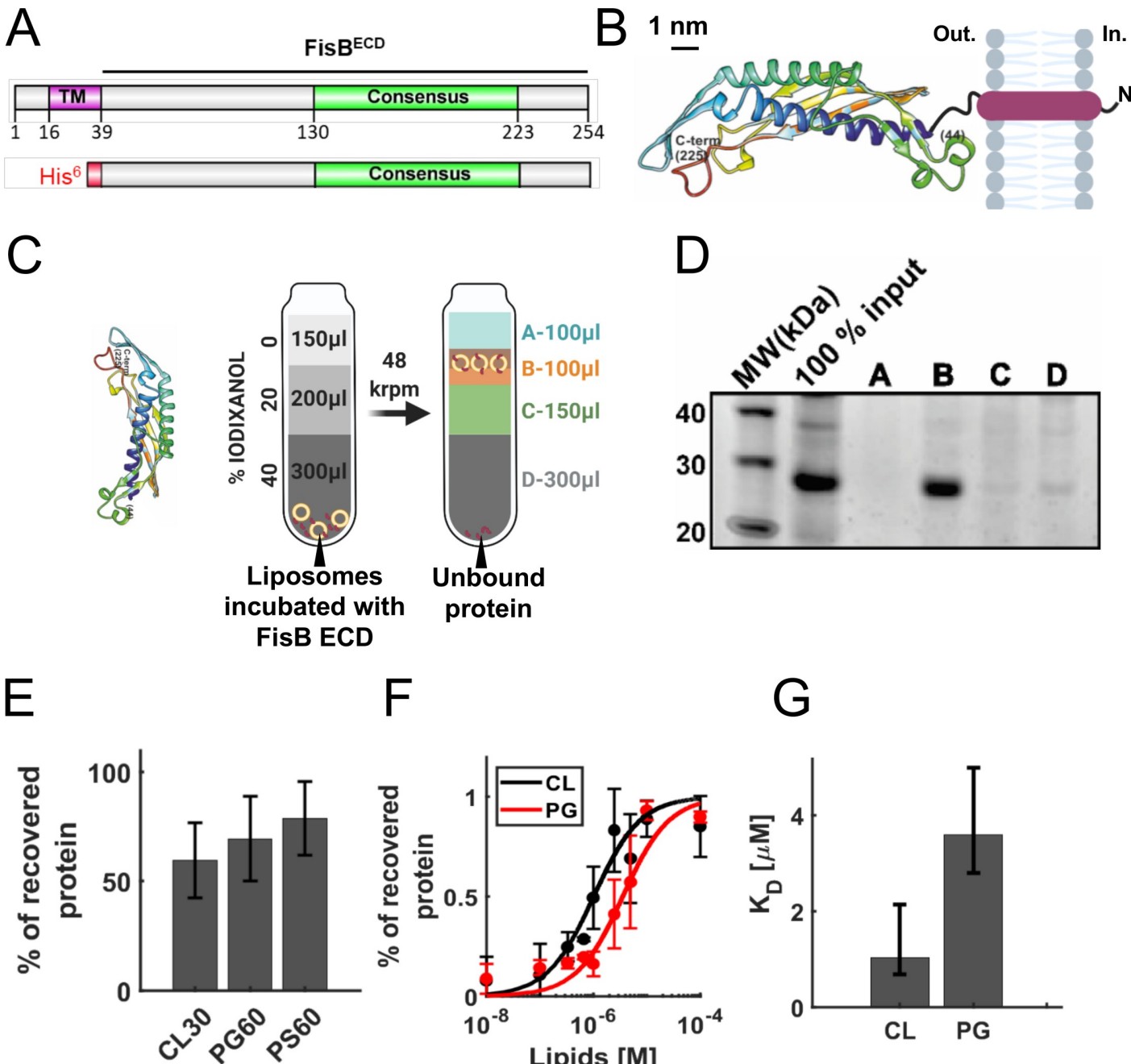

**Fig 4. Binding of FisB ECD to acidic lipids. (A)** Domain structure of FisB and its His6-tagged ECD used in floatation experiments. **(B)** Predicted model of FisB[44-225] comprising most of the ECD [49], schematically attached to the membrane. **(C)** Schematic of the floatation assay. Liposomes (40 nmol total lipid) and FisB ECD (200 pmol) were incubated for 1 hour (total volume of 100 μl) at room temperature and layered at the bottom of an iodixanol density gradient. Upon ultracentrifugation, liposomes float to the top interface, whereas unbound protein remains at the bottom. Four fractions were collected as indicated and analyzed by SDS-PAGE. **(D)** SYPRO orange stained gel of FisB ECD incubated with liposomes containing 45 mole % CL. The percentage of recovered protein is determined by comparing the intensity of the band in fraction B to the input band intensity. **(E)** Indistinguishable amounts of FisB ECD are recovered when FisB ECD is incubated with liposomes containing different acidic lipid species as long as the charge density is similar. CL30, PG60, and PS60 indicate liposomes containing 30 mole % CL, 60 mole % PG, and 60 mole % PS, respectively. CL carries 2 negative charges, whereas PG and PS carry one each. The rest of the liposome composition is PC. **(F)** Fraction of liposome-bound iFluor555-FisB ECD (100 nM) recovered after floatation as a function of lipid concentration. Titration curves were fit to $f_b = K[L]/(1+K[L])$, where $f_b$ is the bound fraction of protein, [L] is the total lipid concentration (assumed to be $\gg$ [protein bound]) and $K = 1/K_d$ the apparent association constant, and $K_d$ is the apparent dissociation constant. **(G)** Best fit values for $K_d$ were 1.0 μM for CL (95% confidence interval, CI = 0.7–2.1 μM) and 3.6 μM for PG (CI = 2.8–5.0 μM), respectively. iFluor555-FisB ECD (100 nM) was incubated with $10^{-8}$ to $10^{-4}$ M lipids for 1 hour at room temperature before flotation. Liposomes contained 45 mole % of CL or PG and 55% PC. CL, cardiolipin; ECD, extracytoplasmic domain; FisB, fission protein B; iFluor555-FisB ECD, iFluor555-labeled FisB ECD; PC, phosphatidylcholine; PG, phosphatidylglycerol; PS, phosphatidylserine.

Together, these results suggest that while FisB has higher affinity for CL than for PG, the higher affinity results mainly from the higher charge carried by CL. FisB does not bind CL with much specificity; at the same surface charge density, FisB ECD binds PG or even PS, which is not a *B. subtilis* lipid, with similar affinity. Thus, in vivo FisB is likely to bind CL as well as PG, which is much more abundant.

## Purified FisB ECD forms soluble oligomers

FisB forms clusters of various sizes in cells as described above (Figs 1 and 2) and does not appear to have other protein interaction partners [23]. Thus, homo-oligomerization of FisB may be important for its function. We explored oligomerization of recombinant, soluble FisB ECD (Fig 5). When FisB ECD bearing a hexa-histidine tag is expressed in *Escherichia coli* and purified to homogeneity by affinity chromatography, samples analyzed by SDS-PAGE show multiple bands corresponding to different oligomeric states (Fig 5D, S1 Appendix Fig I, panel B). Size exclusion chromatography (SEC) analysis resolved the purified protein into predominant high molecular weight oligomeric structures eluting over a wide range of sizes and low molecular weight peaks comprising minor components (Fig 5E, S1 Appendix Fig I, panel C, top). The minor peak at approximately 23 kDa (18-ml elution volume) corresponds to monomeric FisB ECD, whereas the peak at approximately 400 kDa (15 ml) is FisB ECD that co-elutes with another protein, likely the 60-kDa chaperone GroEL, a common contaminant in recombinant proteins purified from *E. coli* (S1 Appendix Fig I, panel D). To rule out potential artifacts caused by the hexa-histidine affinity tag, we also purified FisB ECD using a GST tag, which yielded similar results. The SEC of high molecular weight peaks collected from the initial chromatogram did not show a redistribution when reanalyzed (S1 Appendix Fig I, panel C, bottom), suggesting that once formed, the oligomeric structures are stable for an hour or longer.

We analyzed the high molecular weight SEC fractions (peaks 1 and 2) using electron microscopy (EM) after negative staining. This analysis revealed rod-like structures quite homogeneous in size, approximately 50-nm long and approximately 10-nm wide (Fig 5F, S1 Appendix Fig I, panel E). These structures displayed conformational flexibility, which precluded structural analysis using cryogenic electron microscopy (cryo-EM; and likely hampered our attempts to crystallize FisB ECD). We estimate every rod-like oligomer can accommodate approximately 40 copies of the predicted structure of FisB[44-225] shown in Fig 4B, similar to the number of FisB molecules recruited to the membrane fission site in cells (Fig 2).

## A FisB mutant that is selectively impaired in homo-oligomerization

To determine whether self-oligomerization and lipid binding interactions are important for FisB's function, we generated a series of mutants, characterized oligomerization and lipid binding of the mutant proteins in vitro, and analyzed FisB localization dynamics and membrane fission during sporulation in vivo.

We suspected that self-oligomerization of FisB was at least partially due to hydrophobic interactions. Accordingly, we first mutated conserved residues G175, I176, I195, and I196 in a highly hydrophobic region of FisB ECD (Fig 5A and 5B), producing a quadruple mutant, G175A, I176S, I195T, and I196S (FisB[GIII]). These residues are on the surface of the predicted structure of FisB ECD (Fig 5C), so are not expected to interfere with folding. Purified FisB[GIII] ECD displayed reduced oligomerization when analyzed using SDS-PAGE or SEC (Fig 5D and 5E). Although much reduced in amplitude, a broad, high molecular weight peak was still present in size exclusion chromatograms (Fig 5E). Negative-stain EM analysis of this fraction

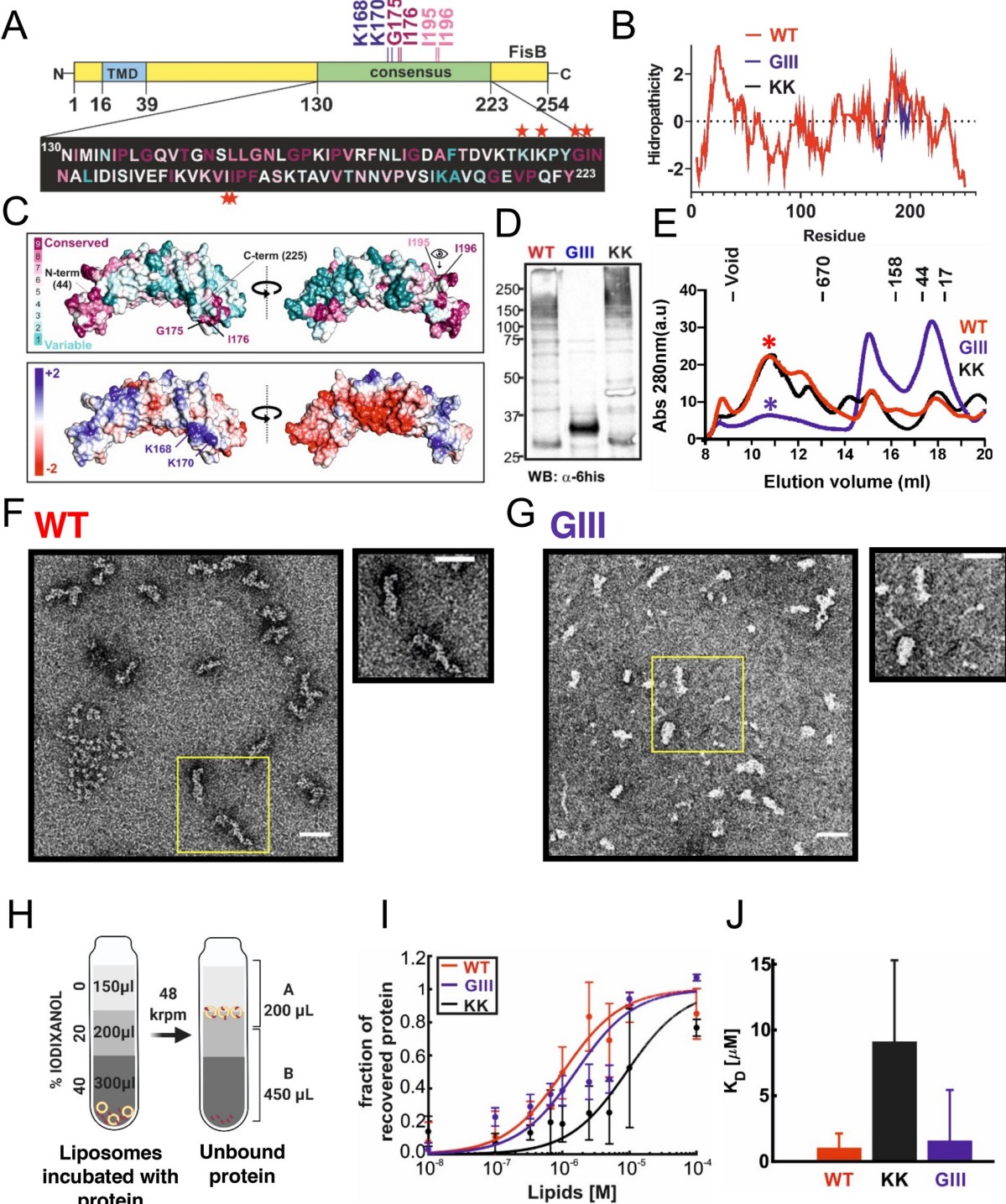

**Fig 5. FisB mutants selectively impaired in oligomerization and membrane binding. (A)** Mutated residues shown on the FisB domain structure. **(B)** Kyle-Doolittle hydrophobicity profile of the FisB sequence for WT, FisB K168D, K170E (FisB[KK]), and FisB G175A,I176S, I195T, I196S (FisB[GIII]) mutants. **(C)** Mutations shown on the predicted model [49] of FisB[44-225]. Residue conservation (top) and electrostatic potential (bottom) are mapped onto the structure. **(D)** Western blot of cell lysates from *E. coli* cells expressing FisB-ECD[WT], FisB-ECD[GIII], or FisB-ECD[KK], probed with an anti-histidine antibody. High molecular weight bands in the WT and KK lanes are largely absent in the GIII lane, indicating that FisB[GIII] is less prone to forming oligomers. **(E)** SEC of FisB WT and the GIII and KK mutants. Intensities of high and low molecular weight peaks are reversed for FisB WT and the GIII mutant, whereas the KK mutant has a profile similar to WT. **(F)** A fraction corresponding to the high-molecular peak in E (indicated by *) for FisB WT was collected and imaged using negative-stain EM, which revealed flexible, elongated structures approximately 50 nm × 10 nm. **(G)** A similar analysis for FisB[GIII] revealed more heterogeneous and less

stable structures. Scale bars in F and G are 50 nm. **(H)** Schematic of the floatation experiments to determine the apparent affinity of FisB mutants for liposomes containing acidic lipids. Experiments and analyses were carried out as in Fig 4, except only 2 fractions were collected. iFluor555-FisB ECD (100 nM) was incubated with $10^{-8}$ to $10^{-4}$ M lipids for 1 hour at room temperature before floatation. Liposomes contained 45 mole % of CL and 55% PC. **(I)** Fraction of protein bound to liposomes as a function of total lipid concentration. Data were fitted to a model as in Fig 4F. The data and fit for FisB WT are copied from Fig 4F for comparison. **(J)** Best fit values for $K_d$ were 1.0 μM for WT (95% confidence interval, CI = 0.7–2.1 μM), 9.1 μM for KK (CI = 6.5–15.3 μM), and 1.6 for GIII (CI = 0.9–5.1 μM), respectively. EM, electron microscopy; FisB, fission protein B; PC, phosphatidylcholine; SEC, size exclusion chromatography; WT, wild-type.

revealed oligomerization with less defined size and structure compared to wild-type FisB ECD (Fig 5G).

To test whether lipid binding of the GIII mutant was affected, we used the co-flotation assay described above, except only 2 fractions were collected (Fig 5H and 5I). This analysis revealed that, despite being impaired in self-oligomerization, FisB$^{GIII}$ ECD has lipid binding properties similar to wild-type with a dissociation constant $K_d^{GIII} = 1.6$ μM (95% confidence interval CI = 0.9 to 5.1 μM), indistinguishable from that of wild-type FisB ECD$^{WT}$ ($K_d^{wt} = 1.0\ \mu M,\ CI = 0.7 - 2.1\ \mu M$, Fig 5J).

## FisB$^{K168D,K170E}$ (FisB$^{KK}$) is selectively impaired in binding acidic lipids

To engineer lipid-binding mutants, we took advantage of our observation that FisB binding to anionic lipids is principally mediated through electrostatic interactions (S1 Appendix Fig I, panel H). We generated a series of mutants in which we either neutralized or inverted up to 4 charges (S1 Appendix Fig K, S1 Appendix Table B). The ECD of a set of charge neutralization mutants was expressed in *E. coli*, purified, and tested for lipid binding using the liposome co-floatation assay. The largest reductions in lipid binding were observed when lysines in a region comprising residues 168 to 172 were neutralized (S1 Appendix Fig K, panel A). This region corresponds to a highly positively charged pocket in the predicted model of FisB 44–225 (Fig 5C).

A partially overlapping set of FisB mutants were expressed in a *ΔfisB* background and tested for sporulation efficiency by monitoring formation of heat-resistant colonies (S1 Appendix Fig K, panels B–E). Again, the strongest reductions in sporulation efficiency were found when lysines 168, 170, or 172 were mutated (S1 Appendix Fig K, panel D). We decided to characterize the K168D, K170E mutation in more detail, as it produced the strongest reduction in sporulation efficiency.

We purified the ECD of FisB$^{K168D,K170E}$ (FisB$^{KK}$) from *E. coli* and tested its binding to liposomes containing 45 mole % CL using the co-floatation assay (Fig 5H–5J). The dissociation constant for FisB$^{KK}$-acidic lipid binding was $K_d^{KK} = 9.1$ μM (CI = 6.5–15.3 μM), nearly 10-fold lower than that for wild-type FisB ECD ($K_d^{wt} = 1.0\ \mu M,\ CI = 0.7 - 2.1\ \mu M$, Fig 5I and 5J). Importantly, formation of oligomers was not affected (Fig 5D and 5E). Thus, FisB$^{KK}$ is specifically impaired in binding to acidic lipids.

## FisB–lipid interactions and homo-oligomerization are important for targeting FisB to the fission site

Using the FisB mutants selectively impaired in binding to lipids or homo-oligomerization, we investigated whether these activities are important for FisB's function in vivo. To analyze FisB clustering and targeting to the fission site, we fused wild-type FisB or the 2 mutants to an N-terminal monomeric YFP (mYFP) and expressed the fusions at lower levels, which facilitated observation of ISEPs (Fig 6A). We induced these strains to sporulate and monitored FisB dynamics and membrane fission. Both the lipid-binding (FisB$^{KK}$) and the oligomerization

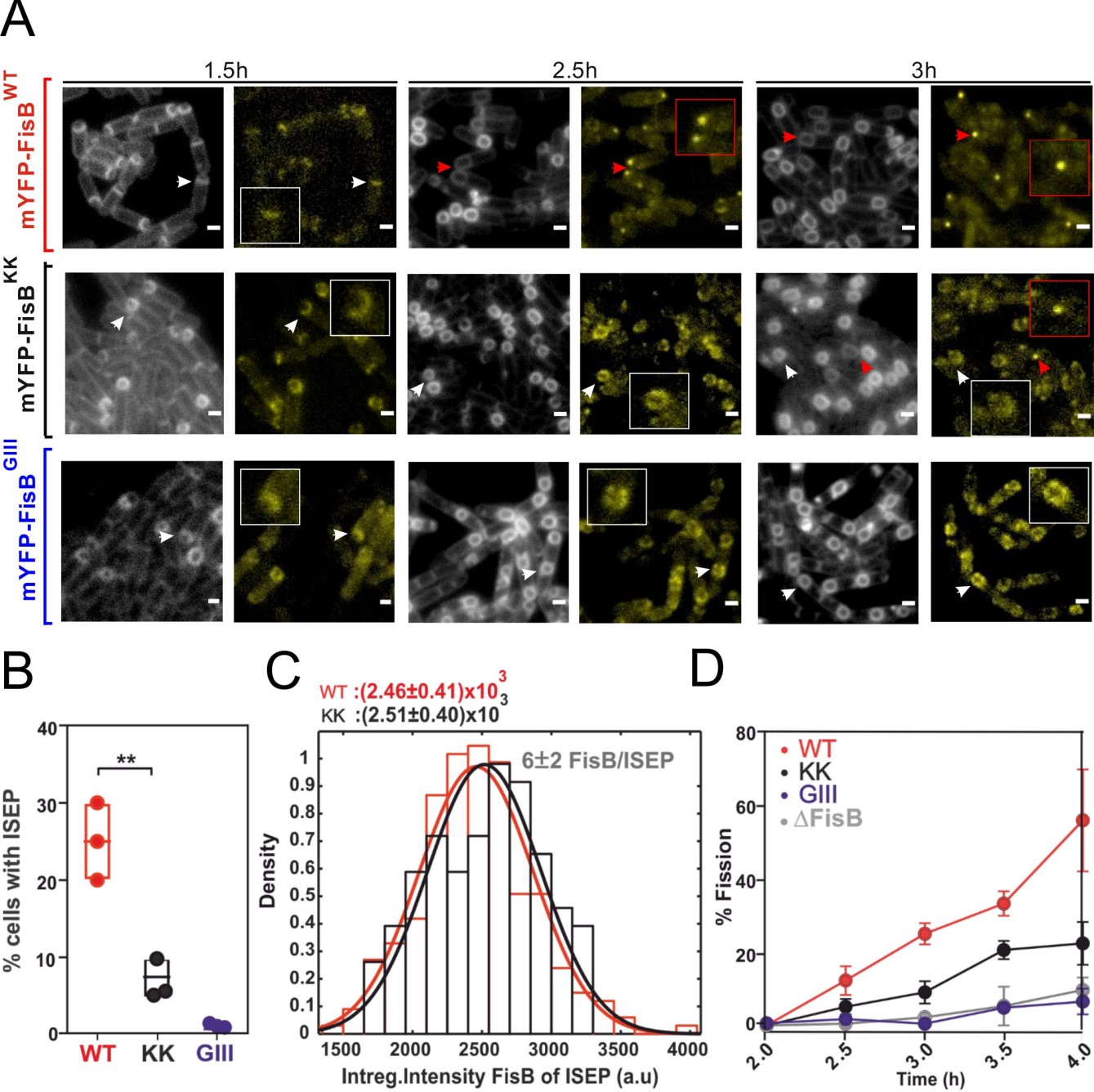

**Fig 6. FisB clustering and binding to acidic lipids are both required for ISEP formation and membrane fission. (A)** Snapshots of sporulating ΔfisB cells expressing mYPF-FisB$^{WT}$ (BAL002), mYPF-FisB$^{KK}$ (BAL006), or mYPF-FisB$^{GIII}$ (BAL007) at low levels. For each time point after downshifting to the sporulation medium, cell membranes were labeled with TMA-DPH, and images were taken both in the membrane (left) and the YFP (right) channels. By t = 2.5 hours, some foci at the engulfment pole (ISEP) are visible for WT cells that have undergone membrane fission (red boxes), but not for the KK or GIII mutants (white boxes). A small fraction of KK mutants (7.3%) accumulated FisB at the engulfment pole and underwent membrane fission at t = 3 hours. Scale bars represent 1 μm. **(B)** Percentage of cells with an intense spot at the engulfment membrane (ISEP) at t = 3 hours into sporulation, for WT FisB, FisB$^{KK}$, or FisB$^{GIII}$. For every strain, 200–300 cells from 3 independent experiments were analyzed at the indicated times during sporulation. **(C)** Distribution of background-corrected integrated intensities (sum of pixel values) of ISEP fluorescence for ΔfisB cells expressing mYPF-FisB$^{WT}$ or mYPF-FisB$^{KK}$. The distributions are indistinguishable. Since low-expression cells accumulate, on average, 6 ± 2 FisB$^{WT}$ molecules at the ISEP (S4D Fig), so do FisB$^{KK}$ cells. A total of 175 and 68 ISEPs were analyzed for WT and KK mutant strains. **(D)** Percentage of cells that have undergone membrane fission at the indicated time points. (For every strain, 200–300 cells from 3 independent experiments were analyzed at the indicated times during sporulation.). FisB, fission protein B; ISEP, intense spot at the engulfment pole; mYFP, monomeric YFP; WT, wild-type.

mutant (FisB$^{GIII}$) were targeted to the cell membrane, unlike many other mutants we tested (S1 Appendix Fig K, panel E, S1 Appendix Table B). At t = 1.5 hours after the nutrient downshift, mYFP-FisB signals were visible in all strains without any distinguishing features. At t = 2.5 hours, a subset of cells expressing the wild-type FisB fusion had undergone membrane fission, and these cells had an ISEP. By contrast, membrane fission was not evident in either of the mutants. By t = 3 hours, 25% of WT FisB cells had undergone fission, nearly always with an accompanying ISEP. In the lipid binding FisB$^{KK}$ mutant, only 8% of the sporulating cells had accomplished membrane fission (Fig 6B), but more than 90% of those that did had an ISEP (53/58 cells). Membrane fission events and the accompanying bright mYFP-FisB spots were very rare (0.6%) in the oligomerization-deficient FisB$^{GIII}$ mutant.

The distribution of fluorescence intensities of the foci from low-expression WT and KK cells were indistinguishable (Fig 6C). Using the DNA origami fluorescence intensity calibration (Fig 2), we estimate 6 ± 2 copies of low-expression FisB WT or the KK mutant to have accumulated at the fission site. For the GIII mutant, there were not enough cells with an intense spot to perform a similar analysis.

From TMA-DPH labeling, we determined the fraction of cells that successfully completed fission as a function of time (Fig 6D). Oligomerization-deficient FisB$^{GIII}$ was not able to induce fission, whereas the lipid-binding mutant FisB$^{KK}$ had a partial, but severe defect (approximately 50% reduction compared to wild type). Importantly, both mutants were expressed at levels similar to the wild type (S1 Appendix Fig J), so the defects to form an ISEP and undergo membrane fission are not due to lower expression levels.

Together, these results suggest that FisB–lipid and FisB–FisB interactions are both important for targeting FisB to the fission site.

## *C. perfringens* FisB can substitute for *B. subtilis* FisB

So far, our results suggest that FisB–FisB and FisB–acidic lipid interactions are the main drivers for targeting FisB to the membrane fission site. If no other partners are involved, FisB should be largely an independent fission module, i.e., FisB homologs from different sporulating bacteria should be able to substitute for one another at least partially, even if sequence homology is low outside the consensus region. To test this idea, we expressed *C. perfringens* FisB (FisB$^{Cperf}$) in *B. subtilis* cells lacking FisB (BAL005). The sequence identity is only 23% between FisB sequences from these 2 species. In the heat-kill assay, FisB$^{Cperf}$ fully rescued *B. subtilis ΔfisB* defects (Fig 7A). *C. perfringens* FisB fused to mEGFP (mEGFP-FisB$^{Cperf}$) had similar dynamics as FisB$^{Bsubti}$, forming DMCs at early times that gave way to an ISEP where membrane fission occurs (Fig 7B). Population kinetics of membrane fission were slower with FisB$^{Cperf}$ (Fig 7C), but nearly every cell that underwent fission had an ISEP as for the wild-type protein (220/239, or 92%). The intensity distribution of mEGFP-FisB$^{Cperf}$ ISEP was shifted to smaller values compared to mEGFP-FisB$^{Bsubti}$ ISEP (Fig 7D). Since the average ISEP intensity for FisB$^{Bsubti}$ corresponds to approximately 40 copies (Fig 2), we deduce approximately 9 copies of FisB$^{Cperf}$ accumulate at ISEP at the time of membrane fission. At t = 3 hours into sporulation, the percentage of cells with an ISEP was lower for cells expressing mEGFP-FisB$^{Cperf}$ (Fig 7E).

In all conditions tested so far, nearly all cells that had undergone membrane fission also had an intense FisB spot at the engulfment pole (Figs 2, 3, 6, and 7). When we plotted the percentage of cells having an ISEP against the percentage of cells that have undergone fission at t = 3 hours, we found a nearly perfect correlation (Fig 7F). FisB$^{Cperf}$ fit this pattern well, despite having a low sequence identity to FisB$^{Bsubti}$, suggesting a common localization and membrane fission mechanism, likely based on a few conserved biophysical properties.

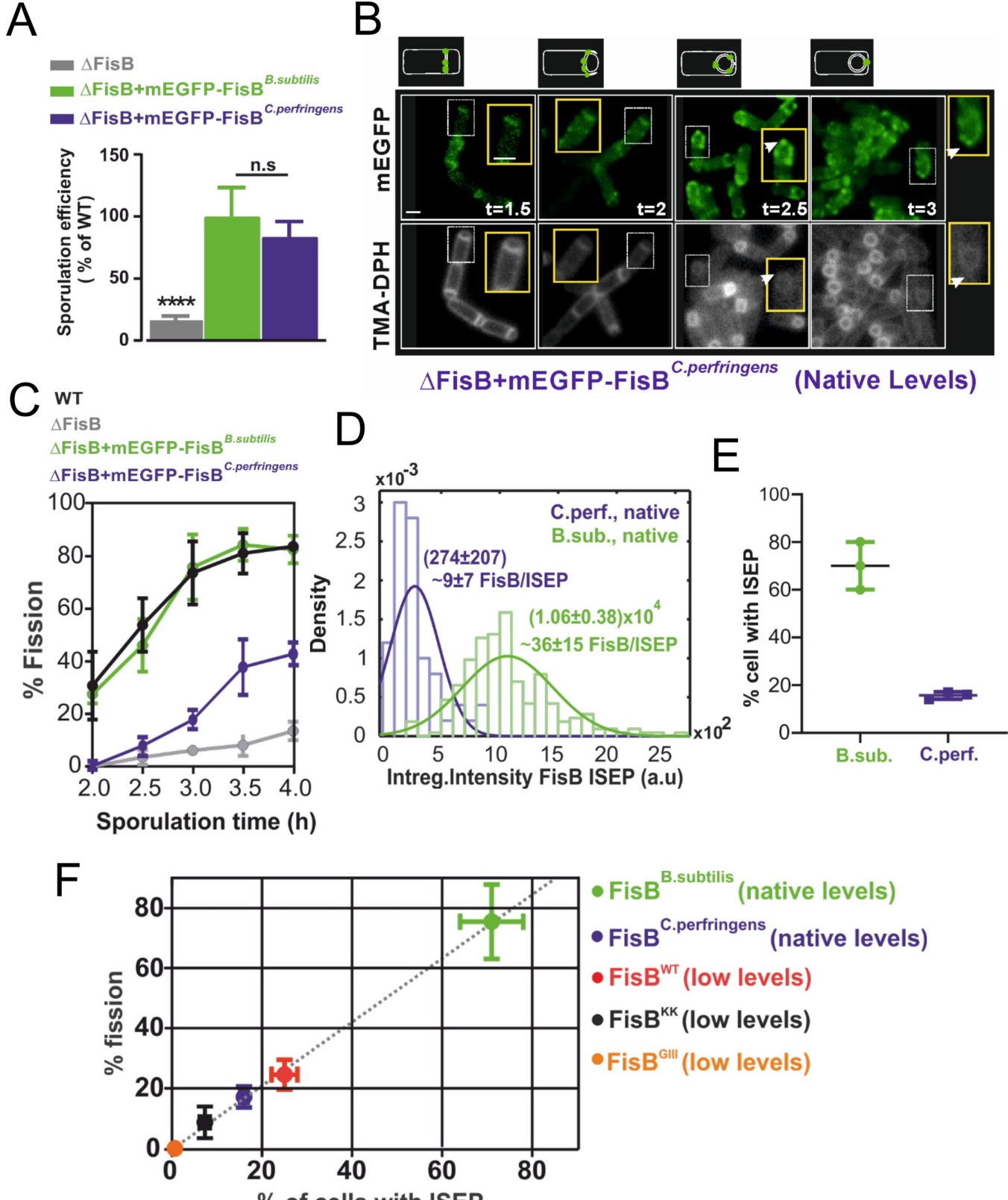

**Fig 7. *C. perfringens* FisB can substitute for *B. subtilis* FisB despite poor sequence identity.** (A) Heat-resistant CFUs for ΔfisB cells expressing *B. subtilis* (BAL001) or *C. perfringens* FisB (BAL005) at native levels, presented as a percentage of the WT sporulation efficiency. Results are shown as means ± SD for 3

replicates per condition. **(B)** Snapshot of ΔfisB cells expressing mEGFP-FisB$^{Cperfringens}$. Aliquots were removed at the indicated times, membranes labeled with TMA-DPH, and both the TMA-DPH and the EGFP channels imaged after mounting into agar pads. White boxed areas are shown on an expanded scale in yellow boxes. Arrows indicate cells with ISEP that have undergone membrane fission. Bar, 1 μm. **(C)** Percentage of cells that have undergone membrane fission as a function of sporulation time for WT cells, ΔfisB cells, ΔfisB cells expressing *B. subtilis* mEGFP-FisB at native levels, or ΔfisB cells expressing mEGFP-FisB$^{Cperfringens}$. The plots for the first 3 conditions are reproduced from Fig 1F for comparison. **(D)** Distribution of background-corrected total fluorescence intensity of ISEP for ΔfisB cells expressing mEGFP-FisB$^{Cperfringens}$ or mEGFP-FisB$^{Bsubtilis}$ at native levels. From the calibration in Fig 2D, we estimate 9 ± 7 FisB$^{Cperfringens}$ per ISEP. The distribution for mEGFP-FisB$^{Bsubtilis}$ is reproduced from Fig 2C for comparison (150 and 93 ISEPs were analyzed for mEGFP-FisB$^{Bsubtilis}$ and mEGFP-FisB$^{Cperfringens}$, respectively.) **(E)** Percentage of cells with ISEP, for ΔfisB cells expressing mEGFP-FisB$^{Cperfringens}$ or mEGFP-FisB$^{Bsubtilis}$. (For each strain, 200–300 cells from 3 independent experiments were analyzed.) **(F)** Percentage of cells that have undergone membrane fission at t = 3 hours vs. the percentage of cells with ISEP at the same time point, for the conditions indicated. There is a nearly perfect correlation between these 2 quantities (the dashed line is a best-fit, y = 1.03x, $R^2$ = 0.96). CFU, colony-forming unit; FisB, fission protein B; ISEP, intense spot at the engulfment pole; WT, wild-type.

## FisB does not have a preference for highly curved membrane regions, but can bridge membranes

A number of proteins localize to subcellular sites due to their preference for curved membrane regions [60–63]. During late stages of engulfment, the most highly curved region in the cell is the membrane neck connecting the engulfment membrane with the rest of the mother cell membrane, and this is where FisB accumulates. We therefore asked whether curvature sensing could be a mechanism driving FisB's localization. To test this possibility, we undertook 3 independent series of experiments.

First, we used the principle that any protein that preferentially binds curved membranes at low membrane coverage can also induce membrane curvature when present at sufficiently high coverage [61,64]. Thus, we tested whether the soluble ECD of FisB could generate curved regions in highly malleable membranes of GUVs at high coverage. We incubated 2 μM purified soluble FisB ECD labeled with iFluor555 with GUVs and monitored protein coverage and membrane deformations using spinning-disc confocal microscopy. Even when the GUV membranes were covered uniformly with iFluor555-FisB ECD, we could not observe any GUV membrane deformations (Fig 8A). As a positive control, we used purified Endophilin A1 (EndoA1, labeled with Atto395), an N-BAR domain containing endocytic protein [65–67]. We incubated 2 μM EndoA1 with GUVs composed of 45% DOPS, 24.5% DOPC, 30% DOPE, and 0.5% DiD, which resulted in extensive tubulation of GUV membranes (Fig 8A), as reported previously [68]. Importantly, the difference in the membrane sculpting ability of the 2 proteins is not due a weaker affinity of FisB ECD for membranes ($K_d$≈1 μM for membranes with 45 mole % CL, Fig 4F) compared to endophilin ($K_d$ = 1.15 μM for membranes containing 45% DOPS, 30% DOPE, 24.5% DOPC, 0.5% TR-DHPE [66]).

Second, we slowly deflated GUVs to facilitate any potential membrane curvature generation by FisB ECD (which works against membrane tension) and/or to provide curved regions to test if FisB ECD accumulated there. Deflated GUVs displayed curved regions because their larger surface-to-volume ratios no longer allowed spherical shapes. Even under these favorable conditions, FisB ECD was not able to generate highly curved regions on these deflated GUVs (Fig 8B). In addition, if FisB ECD had a preference for negatively (positively) curved regions, it should accumulate at such regions while being depleted from positively (negatively) curved areas. Quantification of FisB ECD coverage at negatively or positively curved membrane regions showed no curvature preference (Fig 8B).

Third, we tested if FisB's localization in live *B. subtilis* cells depended on membrane curvature. To avoid potentially confounding effects of other cues that may be present during sporulation, we expressed GFP-FisB under an inducible promoter during vegetative growth. In addition, we blocked cell division by inducing expression of MciZ [70]. MciZ normally blocks binary cell division during sporulation, but when expressed during vegetative growth, cells

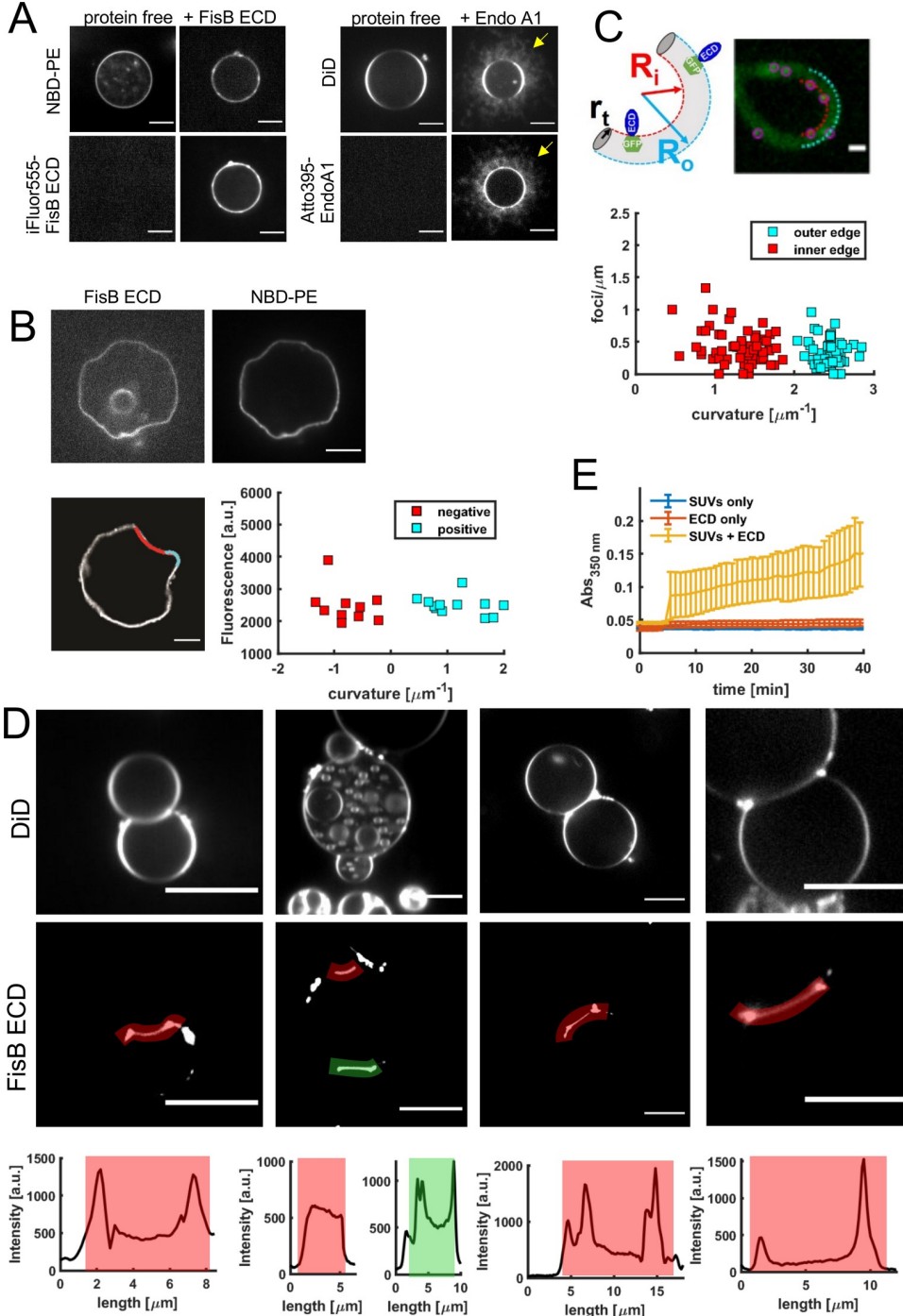

**Fig 8. FisB does not sense or induce membrane curvature. (A)** FisB ECD does not induce deformation of GUV membranes. Left: GUVs incubated with 2 μM iFluor555-FisB ECD did not show any tubulation or invagination of the GUV membrane. GUVs were composed of (in mole %: 25 *E. coli* PE, 5 *E. coli* CL, 50 *E. coli* PG, 19 eggPC, and 1 NBD-PE). Right: incubation of 2 μM endophilin A1 (EndoA1, labeled with Atto395) with GUVs (45% DOPS, 24.5% DOPC, 30% DOPE, and 0.5% DiD) resulted in extensive tubulation of membranes (arrows), as reported previously [22]. The 2 proteins have similar affinities for GUV membranes under these conditions (Fig 4F and [69]). **(B)** FisB ECD cannot deform deflated GUVs and its membrane localization is independent of curvature. To avoid potential issues with high membrane tension preventing membrane deformation, GUVs were deflated using osmotic stress, which resulted in deformed GUVs with both negatively and positively curved regions. FisB ECD bound to these GUVs was unable to induce any high-curvature deformations. The intensity of iFluor555-FisB ECD along a membrane contour (proportional to coverage) was plotted against membrane curvature in the corresponding region. There was

no correlation between membrane curvature and FisB ECD coverage. **(C)** FisB localization does not depend on curvature in filamentous *B. subtilis* cells. GFP-FisB was expressed under an inducible promoter during vegetative growth, and cell division was blocked by inducing expression of MciZ[70]. Cells grew into long flexible filaments that were bent to varying degrees. The linear density of GFP-FisB spots (spots/μm) was independent of filament curvature. **(D)** FisB ECD bridges GUV membranes. iFLuor555-FisB ECD (100 nM) was incubated with GUVs (same composition as in A and B). Many GUVs were found adhering to one another. iFluor555-FisB ECD signals were enhanced in the adhesion patches, in particular at the rims. Intensity profiles along the highlighted contours are shown below the examples. **(E)** FisB ECD aggregates small liposomes. Liposomes (in mole %: 25 *E. coli* PE, 5 *E. coli* CL, 50 *E. coli* PG, 19 eggPC, 50 μM total lipid) were incubated in the absence and presence of FisB ECD (unlabeled) and their aggregation monitored by absorbance at 350 nm. FisB was added at 5 minutes (1 μM final), which caused the absorbance to increase, indicating increased liposome aggregation. CL, cardiolipin; ECD, extracytoplasmic domain; eggPC, egg L-α-phosphatidylcholine; FisB, fission protein B; GUV, giant unilamellar vesicle; iFluor555-FisB ECD, iFluor555-labeled FisB ECD; PE, phosphatidylethanolamine; PG, phosphatidylglycerol; SUV, small unilamellar vesicle.

grow into long flexible filaments that are bent to varying degrees, providing regions with different membrane curvatures. We imaged GFP-FisB spots along curved edges of these filaments and plotted the linear density of GFP-FisB spots (spots/μm) as a function of filament curvature (Fig 8C). There was no clear correlation between GFP-FisB spot density and filament curvature. Although this method generates a limited amount of curvature, a similar approach was previously used to show that DivIVA preferentially localizes to negatively curved regions [71].

In the GUV experiments, we noticed that FisB ECD caused GUVs to adhere to one another when they came into contact, accumulating at the adhesion patch between the membranes and at the rims (Fig 8D). Absorbance measurements using small unilamellar vesicles (SUVs) confirmed that FisB ECD can bridge membranes and aggregate liposomes (Fig 8E).

Overall, these experiments suggest that FisB does not have any intrinsic membrane curvature sensing/sculpting ability, but it can bridge membranes.

## Modeling suggests that self-oligomerization and membrane bridging are sufficient to localize FisB to the membrane neck

To test the hypothesis that the homo-oligomerization, lipid binding, and the unique membrane topology could be sufficient to recruit FisB to the membrane neck, we considered a minimal model based on free energy minimization. As depicted in Fig 9, we consider the free energy $F$ of an axisymmetric cylindrical membrane neck of radius $R$ and length $L$ connecting two planar membrane sheets, corresponding to the local geometry where the engulfment membrane meets the rest of the mother cell membrane. We assume that the surface density $\phi$ of FisB proteins in the neck is uniform and reaches equilibrium with a surface density $\phi_0$ of FisB in the surrounding membranes and ask whether the neck geometry alone is enough to account for the observed FisB accumulation. The energy functional consists of a term accounting for membrane bending and tension, $F_m$, and another term accounting for FisB protein–protein interactions, $F_P$. We employ the classical Helfrich–Canham theory [72–77] for the energy of the membrane

$$F_m = \int_{S_n} dS_n \left[\frac{\kappa}{2}H^2 + \gamma\right] + \int_{S_S} dS_S \gamma, \tag{1}$$

where $S_n$ and $S_s$ are the surfaces of the membrane neck and sheets, $H$ is twice the mean curvature, $\kappa$ is the bending modulus, and $\gamma$ is the surface tension. The two surrounding membrane sheets are assumed to be planar; thus, their only contribution to the energy comes from membrane tension.

For FisB proteins in the neck, we include translational entropy, the energy of homo-oligomerization in *trans*, and an energy that limits crowding. As shown above, FisB proteins do not

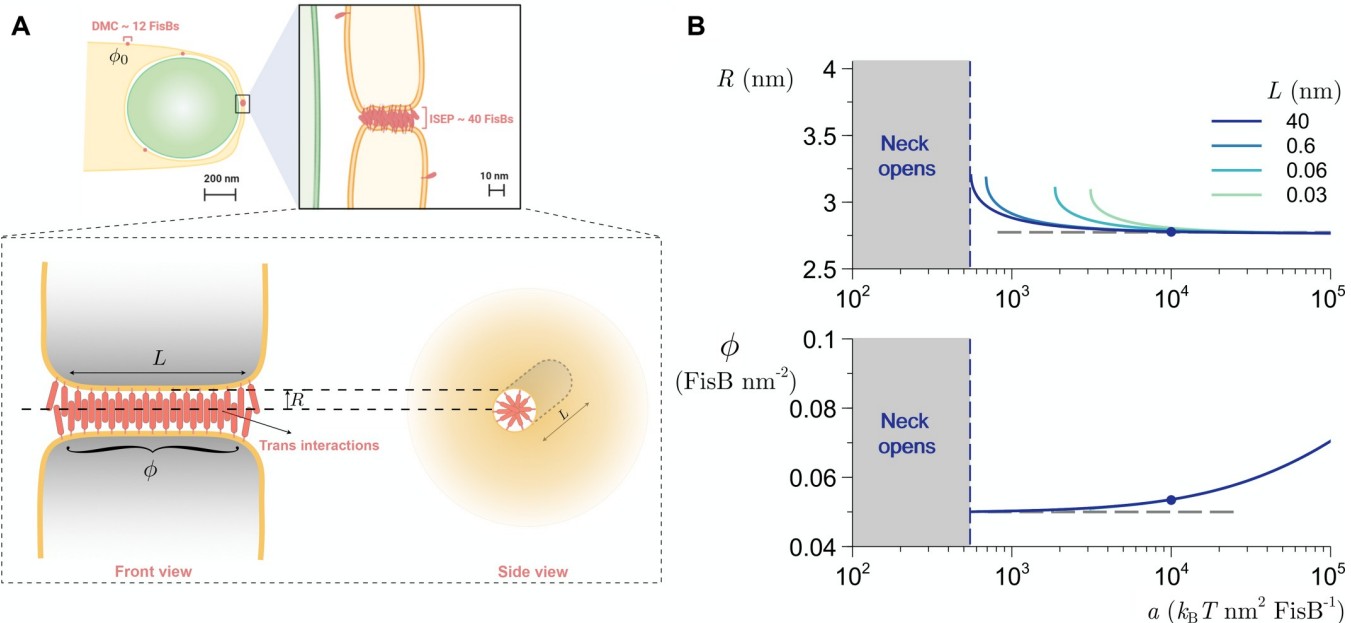

**Fig 9. Modeling supports recruitment of FisB to membrane neck via oligomerization without curvature sensing.** (A) Left: schematic of the late stages of engulfment, when a small membrane neck connects the engulfment membrane to the rest of the mother cell membrane. Right: schematic of FisB accumulation at the fission site. FisB freely moves around the engulfment membrane and other regions of the mother cell membrane, forming clusters of up to approximately 12 molecules. Cluster motions are independent of lipid microdomains, flotillins, the cell wall synthesis machinery, and voltage or pH gradients. About 40 copies of FisB accumulate at the membrane neck in an immobile cluster. Bottom: modeled axisymmetric membrane neck of radius $R$ and length $L$ connecting two membrane sheets. The uniform areal concentration of FisB in the neck is $\phi$. (B) Top: equilibrium radius of the neck as a function of FisB *trans* homo-oligomerization strength, $a$, for several values of neck length, $L$. Below a minimum interaction strength, FisB cannot stabilize the neck and the neck opens. The horizontal line is the radius corresponding to the minimum of the potential describing the *trans*-interaction, $R = 2^{1/6}\sigma$ (Eq 3). Bottom: equilibrium FisB concentration in the neck as a function of $a$. The horizontal line is $\phi_{r_{max}} = 2/(3^{3/2}r_{max}^2)$, the concentration of FisB at the onset of in-plane crowding. Model parameters (see Eqs 5 and 6): $\kappa = 20\, k_B T$ (ref. 78), $\phi_0 = 100$ FisB $\mu m^{-2}$, $\gamma = 10^{-4}$ N m$^{-1}$ (ref. 79), $\sigma \simeq \sigma_{cis} = 2.47$ nm, $\epsilon = 32.78\, k_B T$ nm$^{-2}$, $\phi_{r_{max}} = 5 \times 10^4$ FisB $\mu m^{-2}$, $L = 40$ nm, and for the dot $a = 10^4\, k_B T$ nm$^2$ FisB$^{-1}$. For details, see S2 Appendix. DMC, dim mobile cluster; FisB, fission protein B.

exhibit curvature sensing, so we do not include a term coupling FisB density to membrane curvature in Eq 1. This results in the following expression for the protein free energy [64,80–82]:

$$F_p = \int_{S_n} dS_n \left\{ k_B T \phi \ln\left(\frac{\phi}{\phi_0}\right) + a V_{LJ}(R)\phi^2 + U(\phi) \right\}, \tag{2}$$

where the first term accounts for translational entropy, the second term is an energy per unit area accounting for *trans*-interactions of FisB (where $a$ is the energy per FisB density), and which for simplicity is assumed to be proportional to the standard Lennard–Jones (LJ) potential accounting for a longer-range attraction and shorter-range repulsion,

$$V_{LJ}(r) = \left(\frac{\sigma}{r}\right)^{12} - \left(\frac{\sigma}{r}\right)^6, \tag{3}$$

where $\sigma$ is the length scale at which the *trans*-interaction energy crosses from repulsive to attractive. Finally, the function $U(\phi)$ is an energy penalty for crowding that increases rapidly above a certain FisB concentration. To obtain $U(\phi)$, we assume a purely repulsive, truncated and shifted LJ potential between *cis*-neighboring FisB molecules, which we take to occupy a triangular lattice. Therefore, $U(\phi) = \epsilon[V_{LJ}(r(\phi)) - V_{LJ}(r_{max})]$ when $r \leq r_{max}$ and 0 when $r > r_{max}$, where we have chosen $r_{max} = 2^{1/6}\sigma_{cis}$, namely the minimum of the LJ potential with length

scale $\sigma_{\text{cis}}$. The result is

$$U(\phi) = \epsilon \frac{(\phi^3 - \phi_{r_{\max}}^3)^2}{4\phi_{r_{\max}}^6} \text{ for } \phi \geq \phi_{r_{\max}} \text{ or } 0 \text{ for } \phi < \phi_{r_{\max}}, \tag{4}$$

where $\phi_{r_{\max}} = 2^{\frac{2}{3}}/(3^{\frac{3}{2}}\sigma_{\text{cis}}^2) = 2/(3^{\frac{3}{2}}r_{\max}^2)$ is the FisB concentration corresponding to a nearest neighbor distance $r_{\max}$.

We minimize $F = F_{\text{m}}+F_{\text{p}}$ with respect to $\phi$ to obtain an equation for the equilibrium density of FisB proteins in the neck

$$k_{\text{B}}T\left[1 + \ln\left(\frac{\phi}{\phi_0}\right)\right] + 2a\phi\left[\left(\frac{\sigma}{R}\right)^{12} - \left(\frac{\sigma}{R}\right)^6\right] + \partial_\phi U(\phi) = 0. \tag{5}$$

Then, minimizing $F$ with respect to $R$ yields an equation that determines the equilibrium radius of the neck

$$\gamma_{\text{eff}} - \frac{\kappa}{2R^2} + a\phi^2\left[6\left(\frac{\sigma}{R}\right)^6 - 12\left(\frac{\sigma}{R}\right)^{12}\right] - \frac{2\gamma R}{L} = 0, \tag{6}$$

where $\gamma_{\text{eff}} = \gamma + k_B T\phi\ln\left(\frac{\phi}{\phi_0}\right) + U(\phi)$.

It is important to emphasize that thermal fluctuations and hydrodynamics may play an important role in membrane fission dynamics. However, our intention here is specifically to model the formation of a stable FisB cluster in the neck prior to fission; thus, we neglect these additional effects which are beyond the scope of the present work.

It is also important to point out that the equilibrium radius of the neck in the presence of FisB is significantly smaller than the observed length of the neck. Hence, we expect that, due to the slenderness of the neck, boundary effects will not play a major role. For this reason, we have neglected small contributions from the boundaries in the minimization problem described above (see S2 Appendix for a more detailed discussion).

Fig 9B shows $R$ and $\phi$ as functions of the FisB trans homo-oligomerization strength $a$, for different values of surface tension $\gamma$ and neck length $L$. For realistic parameters (the dot), we find that FisB *trans*-interactions are strong enough to stabilize the neck at $R$ approximately 3 nm, with a close-packed concentration of FisB in the neck $\phi \approx \phi_{r_{\max}}$. For these same parameters, there is a critical lower limit of $a$ below which the FisB interactions are too weak to stabilize the neck, so the neck opens, i.e., $R\rightarrow\infty$ in our simple model. Additionally, Fig 9B shows that the shorter the length of the neck $L$, the stronger the *trans*-interactions needed to stabilize the neck at a finite radius. This makes intuitive sense: the longer the neck, the more FisB can be present to hold the neck together in opposition to membrane tension. (Note that expanding the radius of the neck actually decreases the total membrane area, which is the sum of the membrane in the neck and in the parallel sheets, so that surface tension tends to make the neck expand—see S2 Appendix).

While the above results suggest that an accumulation of FisB at the neck can be energetically stable, one question is how long it might take to reach that state? We expect nucleation of a critical cluster of FisB to be rate limiting, since the time required for diffusion and capture to reach approximately 40 FisB in the neck is quite short, being approximately 3.9 seconds (see S2 Appendix). To obtain a simple estimate of the nucleation time for both low-expression and native-expression strains, we assume that FisB proteins diffuse independently on the entire membrane and that nucleation of a stable cluster in the neck occurs when $n$ proteins happen to be in the neck at the same time. To this end, we need to estimate the fraction of time there are $n$ or more FisB in the neck, as well as the correlation time, i.e., the time between

uncorrelated samples. Since we assume FisB proteins are independent, the number of proteins in the neck will be Poisson distributed, so we only need to know the average in the neck to obtain the full distribution. The average number of FisB in the neck is its area, $2\pi RL$, times the background concentration, $\phi_0$. Furthermore, the correlation time is simply the time for a FisB to diffuse the length of the neck $L^2/D$. Using $\phi_0 \simeq 20$ FisB $\mu m^{-2}$ (see "About 40 FisB molecules accumulate at the engulfment pole to mediate membrane fission" above) for the low-expression strain yields $\langle FisB \rangle \simeq 0.03$ in the neck. Assuming that the 1-hour delay in membrane fission during sporulation of low-expression strain is due to the time for nucleation, we can infer that the number of FisB proteins required for nucleation is $n \approx 3$ (see S2 Appendix). If the native-expression strain also needs $n \approx 3$ to nucleate, we can estimate its corresponding nucleation time using $\phi_0 \simeq 100$ FisB $\mu m^{-2}$, which yields $\langle FisB \rangle \simeq 0.15$ and a nucleation time of approximately 30 seconds. We conclude that for native expression levels of FisB, nucleation of a stable cluster of FisB at the neck is not likely to be rate limiting for the process of membrane fission.

## Discussion

Previously, we showed that FisB is required for the membrane fission event that marks the completion of engulfment of the forespore by the mother cell [23]. Here, we found that a cluster of FisB molecules is nearly always present at the membrane fission site as evidenced by an ISEP using fluorescently tagged FisB. The number of FisB molecules accumulated at the ISEP correlates well with the fraction of cells having undergone membrane fission at a given time point after induction of sporulation (Figs 1 and 7). In addition, the number of wild-type FisB molecules per ISEP correlates with the total number of FisB molecules per cell (S1 Appendix Fig D). Thus, the kinetics of membrane fission are determined by the accumulation of FisB molecules at the fission site. Lowering FisB expression could slow membrane fission by slowing the accumulation of FisB at the pole or by reducing the number of FisB molecules driving fission after they are localized at the fission site. Our modeling results are consistent with slower ISEP nucleation in the low-expression strain; however, currently, we cannot experimentally distinguish between the 2 possibilities, and both may be operating simultaneously.

How is FisB recruited to the fission site? Our results suggest that FisB does not rely on existing landmarks, lipid microdomains, cell wall remodeling machinery, pH or voltage gradients across the cell membrane, or membrane curvature cues for its dynamic localization. In addition, we could not detect proteins interacting with FisB other than itself using an anti-GFP resin pulling on YFP-FisB [23]. By contrast, we found self-oligomerization and binding to acidic lipids to be critical for FisB's function, and purified FisB ECD can bridge artificial membranes. Together, these results suggest that FisB–FisB and FisB–lipid interactions are key drivers for FisB clustering and accumulation at the membrane fission site.

Can FisB oligomerization and lipid binding be sufficient to accumulate an immobile cluster of FisB molecules at the engulfment pole? Modeling suggests that this is indeed the case. First, the narrow neck enables FisB's on opposing membranes to come close enough to interact in *trans*. We infer this to be the preferred orientation for FisB–FisB interaction, since otherwise large clusters would be expected to form elsewhere as well. Second, the unique geometry of the neck connecting the engulfment membrane to the rest of the mother cell membrane plays an important role, as this is the only region in the cell where a cluster of FisB molecules can be "trapped," i.e., once a cluster is formed inside the neck, it cannot diffuse away without breaking apart. This idea is supported by the fact that we do not observe any FisB accumulation at the leading edge of the engulfment membrane until a thin neck has formed at the end of engulfment.

The first FisB oligomers that appear during sporulation are DMCs, each containing about a dozen FisB molecules. (One possibility is that the DMCs may correspond to local membrane folds stabilized by FisBs interacting in *trans*.) Diffusion of DMCs appears to be Brownian on the 10- to 20-second timescale (Fig 2), although a rigorous analysis would require taking into account the geometry of the system. A DMC can diffuse a typical distance of approximately 1 μm in approximately 5 minutes ($D_{DMC} \approx 3 \times 10^{-3} \mu m^2$/s, Fig 2E). By comparison, engulfment in individual cells takes approximately 60 minutes on average [83]. Although the engulfment time is much longer than the DMC diffusion time, the neck region, with an inner diameter of several nanometers, only forms at the very end of the engulfment process. Thus, approximately 40 FisB molecules could be recruited to the neck through diffusion-limited capture of a few DMCs. However, we could not image such capture events directly and cannot rule out that FisB can also diffuse as monomers and could be recruited to the neck in that form. Indeed, a simple model of the rate of nucleation of a cluster of FisBs at the neck suggests that as few as 3 FisBs interacting in *trans* could be sufficient to form a stable cluster there, with a nucleation time significantly shorter than the engulfment time at native expression levels.

How many FisB molecules are needed for efficient membrane fission? In cells completely lacking FisB, approximately 5% of the cells undergo membrane fission by t = 3 hours, compared to approximately 80% or approximately 30% for cells expressing FisB at native or approximately 8-fold reduced levels, respectively (Fig 1F). The former achieve fission with approximately 40 copies, while the latter with only approximately 6. Thus, FisB is not absolutely required for membrane fission, but it makes it much more efficient, i.e., FisB catalyzes membrane fission. The variable stoichiometry suggests that FisB does not oligomerize into a specific quaternary structure with a definite stoichiometry. This variability appears to be a common property among proteins catalyzing membrane fusion and fission, such as soluble N-ethylmaleimide-sensitive-factor attachment protein receptors (SNAREs) [84–86] or dynamin [14]. The smallest clusters associated with membrane fission had approximately 6 FisB copies on average. This number is likely sufficient to form at least 1 ring inside the membrane neck that eventually undergoes fission. Given that fission can occur in the absence of FisB, it is likely that the FisB cluster cooperates with other cellular processes to produce stress on this membrane neck.

We found that FisB dynamics and membrane fission are not affected by removal of CL, PE, or both. CL and PE are widely implicated in membrane fission and fusion reactions due to their tendency to form non-bilayer structures [50,87–90]. The fact that CL or PE do not affect membrane fission during sporulation is remarkable, because such lipids usually affect the kinetics and/or the extent of fusion/fission reactions even if they are not absolutely required [88]. We tested the role of CL in a strain that lacked all 3 known CL synthases, with no detectable CL levels. A previous study reported that in *ΔclsABC B. subtilis* cells, CL levels increase from undetectable during vegetative growth to readily detectable during sporulation [31], suggesting that a yet unidentified sporulation-specific CL synthase may exist. Our results differ from those of Kawai and colleagues in that we were unable to detect any CL in *ΔclsABC B. subtilis* cells during vegetative growth or sporulation. We suggest that the differences may be due to the different strain backgrounds used [91], PY79 [92] here versus BS168 [93] in Kawai and colleagues or differences in detection sensitivities.

Overall, our results suggest that FisB localizes to the membrane fission site using only lipid binding, homo-oligomerization, and the unique geometry encountered at the end of engulfment. We propose that accumulation of a high enough density of FisB leads to membrane fission, possibly by generating increased stress in the FisB network-membrane composite, or in cooperation with another cellular process. A FisB homologue with low sequence identity partially rescued fission defects in *ΔfisB B. subtilis* cells, consistent with the idea that FisB acts as an independent module relying mainly on homo-oligomerization, lipid binding, and sporulation geometry.

## Materials and methods

### Materials

*E. coli* CL, *E. coli* L-α-PG, egg L-α-phosphatidylcholine (eggPC), *E.coli* L-α-PE, 1,2-dioleoyl-sn-glycero-3-phosphoethanolamine-N-(7-nitro-2-1,3-benzoxadiazol-4-yl) (NBD-PE), 1,2-dioleoyl-sn-glycero-3-phosphoethanolamine (DOPE), 1,2-dioleoyl-sn-glycero-3-phosphocholine (DOPC), and 1,2-dioleoyl-sn-glycero-3-phospho-L-serine (DOPS) were purchased from Avanti Polar Lipids (Alabaster, AL, USA). 1-(4-Trimethylammoniumphenyl)-6-Phenyl-1,3,5-Hexatriene *p*-Toluenesulfonate (TMA-DPH) and *N*-(3-Triethylammoniumpropyl)-4-(6-(4-(Diethylamino) Phenyl) Hexatrienyl) Pyridinium Dibromide (FM4-64), and 1,1′-Dioctadecyl-3,3,3′,3′-Tetramethylindodicarbocyanine (DiD) were from Thermo Fisher Scientific (Waltham, MA, USA). Molybdenum Blue spray reagent was from Sigma-Aldrich (Saint Louis, MI, USA). Carbonyl cyanide m-chlorophenyl hydrazone (CCCP) was purchased from Abcam (Branford, CT, USA), and valinomycin was purchased from VWR International (Radnor, Pennsylvania, USA). 3-(N-maleimidylpropionyl)biocytin (MBP) was obtained from Invitrogen (Waltham, MA, USA) and the HRP-conjugated antibody from eBioscience (San Diego, CA, USA). Zaragozic acid was purchased from Sigma-Aldrich. 4-acetamido-4′-maleimidylstilbene-2,2′-disulfonic acid (AMS) and zaragozic acid were from obtained from Cayman Chemical (Ann Arbor, MI, USA).

### General *B. subtilis* methods

*B. subtilis* strains were derived from the prototrophic strain PY79 [92]. Sporulation was induced in liquid medium at 37˚C by nutrient exhaustion in supplemented DS medium (DSM) [94] or by resuspension according to the method of Sterlini and Mandelstam [95]. Sporulation efficiency was determined in 24 to 30 hours cultures as the total number of heat-resistant (80˚C for 20 minutes) CFUs compared to wild-type heat-resistant CFUs. Lipid synthesis mutants were from the *Bacillus* knock-out collection [96] and all were backcrossed twice into *B. subtilis* PY79 before assaying and prior to antibiotic cassette removal. Antibiotic cassette removal was performed using the temperature-sensitive plasmid pDR244 that constitutively expresses Cre recombinase [96]. Cassette removal was further confirmed by PCR with primers flanking the deletion. *B. subtilis* strains were constructed using plasmidic or genomic DNA and a 1-step competence method. Site-directed mutagenesis was performed using Agilent's (Lexington, MA, USA) Quick-change Lightning kit following manufacturer's instructions, and mutations were confirmed by sequencing. The strains and plasmids used in this study are listed in S1 Appendix Tables B and C, respectively.

### Live-cell fluorescence microscopy of *B. subtilis*

Cells were mounted on a 2% agarose pad containing resuspension medium using a gene frame (Bio-Rad, Hercules, CA, USA). Cells were concentrated by centrifugation (3,300 g for 30 seconds) prior to mounting and visualization. This step had no impact on the localization of the fusion proteins. Fluorescence microscopy was performed using a Leica (Buffalo Grove, IL, USA) DMi8 wide-field inverted microscope equipped with an HC PL APO 100×DIC objective (NA = 1.40) and an iXon Ultra 888 EMCCD Camera from Andor Technology (Belfast, Northern Ireland). Membranes were stained with TMA-DPH at a final concentration of 100 μM. Excitation light intensity was set to 50%, and exposure times were 300 ms for TMA-DPH ($\lambda_{ex}$ = 395/25 nm; $\lambda_{em}$ = 460/50 nm); 500 ms for m(E)GFP ($\lambda_{ex}$ = 470/40; $\lambda_{em}$ = 500 to 550); and 1 second for mYFP ($\lambda_{ex}$ = 510/25; $\lambda_{em}$>530) respectively. Images were acquired with Leica

Application Suite X (LAS X), and analysis and processing were performed using the ImageJ software [97].

## Determination of FisB's topology

We used the substituted cysteine accessibility method (SCAM [98]) to determine the topology of FisB. We first generated stains expressing FisB versions with a single cysteine substitution at position G6, L137, or A245, in a $\Delta fisB$ background. FisB does not have any endogenous cysteines. These point mutations decreased the sporulation efficiency slightly (S1 Appendix Table B), we assume without affecting the topology. We selectively biotinylated extra- or intracellular cysteines of *B. subtilis* protoplasts, produced by addition of 0.5 mg/ml lysozyme and incubating cells at 37˚C for 1 hour with gentle rocking. Protoplasts were then incubated with the membrane-impermeant reagent 3-(N-maleimidylpropionyl)biocytin (MBP). To selectively label extracellular cysteines, protoplasts of sporulating cells at 2.5 hours into sporulation were incubated with 100 μM MPB. The reaction was quenched with 50 mM DTT before cells were lysed with hypotonic shock. To label intracellular cysteines selectively, extracellular cysteines of protoplasts were first blocked AMS before cells were lysed and incubated with 100 μM MPB. The reaction was quenched by addition of 100 μM MPB. FisB was pulled down from the cell lysates as described in [98] using an anti-Myc antibody (mAb #2276), and biotinylated proteins were detected by western blot using a HRP-conjugated-Avidin antibody. Further details are provided in S1 Appendix.

## Expression, purification, and labeling of recombinant FisB protein

Recombinant soluble FisB ECD was purified as described in [23] but with slight modifications. Briefly, His$_6$-FisB ECD was expressed in *E. coli* BL21 (DE3) from New England Biolabs (Ipswich, MA, USA) and purified using HisPur Ni-NTA Resin from Thermo Fisher Scientific. Protein expression was induced with 1 mM IPTG at $OD_{600}$ = 0.6 overnight at 16˚C. Cells were harvested by centrifugation, and the pellet was resuspended in Lysis Buffer (20 mM HEPES, 500 mM NaCl, 0.5 mM TCEP, 20 mM Imidazole, 2% glycerol, 20 mM MgCl$_2$) and flash-frozen in liquid nitrogen. Pellets were thawed on ice, and cells were lysed by 5 passes through a high-pressure homogenizer (Avestin EmulsiFlex-C3, Ottawa, Canada). The lysate was spun down at $100,000 \times g$, and the soluble fraction was incubated with HisPur Ni-NTA Resin for 2.5 hours at 4˚C while rotating. The bound protein was washed with Lysis Buffer, Lysis Buffer containing 50 mM, and, finally, 100 mM Imidazole. The protein was eluted in Elution Buffer (20 mM HEPES, 500 mM NaCl, 0.5 mM TCEP, 200 mM Imidazole, 2% glycerol, 20 mM MgCl2). The protein was concentrated using a Vivaspin centrifugal concentrator with a 10-kDa molecular mass cutoff and the concentration determined by Bradford protein assay. The protein was stored at −80˚C.

In experiments with labeled FisB ECD, we used a cysteine mutation, G123C (FisB ECD does not have any endogenous cysteines). After expression and purification as above, iFluor555-maleimide (AAT Bioquest, Sunnyvale, CA, USA)) was reacted with FisB ECD$^{G123C}$ following the manufacturer's instructions. G123 is in a loop that if removed does not interfere with FisB's function (S1 Appendix Fig J).

## Analytical size exclusion chromatography and negative-stain electron microscopy

For SEC analysis, His$_6$-FisB ECD was loaded onto a Superose 6 Increase 10/300 GL column (GE, Chicago, IL, USA) previously equilibrated with 20 mM HEPES, pH 7.5, 500 mM NaCl, 0.5 mM TCEP, 2% glycerol, 20 mM MgCl$_2$, running at a flow rate of 0.5 ml/min at 4˚C. The

column was calibrated with Bio-Rad's Gel Filtration Standards. For negative stain EM analysis, 4 μL of the indicated elution fractions were applied to 200-mesh copper grids coated with approximately 10-nm amorphous carbon film, negatively stained with 2% (wt/vol) uranyl acetate, and air-dried. Images were collected on a FEI Tecnai T12 microscope, with a LaB6 filament operating at 120 kV, and equipped with a Gatan CCD camera.

## Inhibition of cell wall synthesis and analyses of FisB motions

Overnight cultures of GFP-Mbl (BDR2061) or IPTG-induced mGFP-FisB (BMB014) were diluted in CH medium to OD600 = 0.05. Expression of GFP-FisB was induced with 1 mM IPTG for 2 hours at 37˚C. Expression of GFP-Mbl was induced with 10 mM xylose for 30 minutes when BDR2061 reached OD600 = 0.5. For imaging untreated cells, 1 ml of cells was washed twice with 1 ml PBS and finally resuspended in 10 μl PBS. Moreover, 2 μl of cell suspension was spread on a 2% PBS agar pad for imaging. To inhibit cell wall synthesis, 50 μg/ml fosfomycin was added to the cultures 45 minutes before imaging. A total of 1 ml of cells was washed twice with PBS containing 50 μg/ml fosfomycin and mounted on a PBS agar pad also containing fosfomycin. Cells were imaged using an Olympus (Center Valley, PA, USA) IX81 microscope with a home-built polarized TIRF setup [99,100]. Exposure times were 50 ms for BDR2061 and 100 ms for BMB014. Movies were acquired at 1 frame/s. Movies collected for BMB014 were corrected for bleaching using the Bleaching Correction function (exponential method) in ImageJ. Kymographs were created with imageJ along the indicated axes. GFP fusion proteins were tracked using the ImageJ plugin TrackMate [101]. A Laplacian of Gaussian (LoG) filter was used to detect particles with an estimated blob diameter 400 μm. Particles were tracked using the Simple LAP tracker with a 0.25 μm maximum linking distance and no frame gaps. MATLAB (MathWorks, Natick, Massachusetts, USA) was used for further processing of the tracks. MSD was calculated using the MATLAB class @msdanalyzer [102].

The asymmetry of individual tracks (S1 Appendix Fig E, panel F) was calculated as described in [103] using

$$Asym = -log\left(1 - \frac{\left(R_1^2 - R_2^2\right)^2}{\left(R_1^2 + R_2^2\right)^2}\right)$$

where $R_1$ and $R_2$ are the principal components of the radius of gyration, equal to the square roots of the eigenvalues of the radius of gyration tensor $\boldsymbol{R_g}$:

$$\boldsymbol{R_g}(\boldsymbol{i,j}) = \langle \boldsymbol{x_i x_j} \rangle - \langle \boldsymbol{x_i} \rangle \langle \boldsymbol{x_j} \rangle.$$

## Tracking fluorescently labeled FisB spots and estimation of diffusion coefficients

For estimating the mobility of DMC and ISEP, time-lapse movies were recorded with a frame rate of 1 second using wide-field microscopy (50% LED intensity, 300 ms exposure time, gain 300). Spot positions were tracked using SpeckleTrackerJ [48], a plugin for the image analysis software ImageJ [97]. MSDs were calculated using the MATLAB class @msdanalyzer [102].

## Dissipation of membrane potential

Cells were concentrated by centrifugation (3,300 x g for 30 seconds), and 100 μM CCCP or 30 μM valinomycin was added just prior to mounting cells onto a 2% PBS agar pad also containing 100 μM CCCP or 30 μM valinomycin.

## Lipid extraction and thin-layer chromatography (TLC)

Lipids were extracted from *B. subtilis* cells at 3 hours into sporulation according to the method of Lacombe and Lubochinsky [104]. Lipid extracts were analyzed by TLC on silica gel plates in mixtures of chloroform:hexane:methanol: acetic acid (50:30:10:5). Phospholipids were detected with Molybdenum Blue Reagent (Sigma-Aldrich).

## Liposome preparation

SUVs were prepared by mixing 1 μmol of total lipids at desired ratios. A thin lipid film was created using a rotary evaporator (Buchi, New Castle, USA). Any remaining organic solvent was removed by placing the lipid film under high vacuum for 2 hours. The lipid film was hydrated with 1 ml of RB-EDTA buffer [25 mM HEPES at pH 7.4, 140 mM KCl, 1 mM EDTA, 0.2 mM tris(2-carboxyethyl) phosphine] by shaking using an Eppendorf Thermomix for >30 minutes. The lipid suspension was then frozen and thawed 7 times using liquid nitrogen and a 37˚C water bath and subsequently extruded 21 times through a 100-nm pore size polycarbonate filter using a mini-extruder (Avanti Polar Lipids). All SUVs contained 1% NBD-PE to determine the final lipid concentration.

GUVs were prepared by electroformation [105]. Briefly, lipids dissolved in chloroform were mixed in a glass tube at desired ratios and spotted on 2 indium tin oxide (ITO) coated glass slides. Organic solvent was removed by placing the lipid films in a vacuum desiccator for at least 2 hours. A short strip of copper conductive tape was attached to each ITO slide, which were then separated by a 3-mm thick Polytetrafluoroethylene (PTFE) spacer and held together with binder clips. The chamber was filled with 500-μl Swelling Buffer (1 mM HEPES, 0.25 M sucrose, 1 mM DTT) and sealed with Critoseal (VWR International). GUVs were formed by applying a 1.8 V sinusoidal voltage at 10 Hz for at least 2 hours at room temperature.

For experiments involving FisB ECD, the GUVs were composed of (all in mole percentages) 25 *E. coli* PE, 5 *E. coli* CL, 50 *E. coli* PG, 19 eggPC, and 1 DiD or 1 NBD-PE. For experiments in which EndoA1 was used, GUV composition was (all in mole %) 45 DOPS, 24.5 DOPC, 30 DOPE, and 0.5 DiD.

## Liposome–protein co-floatation

For initial experiments, 40-nmol total lipid was incubated with 200 pmol FisB ECD for 1 hour at room temperature in a total volume of 100 μl. A total of 200 μl of 60% Optiprep (iodixanol, Sigma-Aldrich) was added to the sample, creating a 40% Optiprep solution. The sample was then layered at the bottom of a 5 mm × 41 mm Beckman ultracentrifuge tube (#344090) and overlaid with 200 μl of 20% Optiprep and finally 150 μl of buffer (Fig 4C). Liposome-bound proteins co-float to a light density, while unbound proteins pellet upon ultracentrifugation for 1.5 hours at 48 krpm. Fractions were collected as shown in Fig 4C, and the amount of recovered protein was determined by SDS-PAGE (Nu-PAPGE 12% Bis-Tris gel, Thermo Fisher Scientific) stained with SYPRO Orange (Invitrogen).

## Liposome aggregation using absorbance

SUVs were prepared by extrusion as described above but using a 50-nm polycarbonate filter. SUVs were composed of 50 mole % *E. coli* PG, 25 mole % *E. coli* PE, 20 mole % eggPC, and 5 mole % *E. coli* CL. The absorbance at 350 nm of 50-μM total lipid was measured for 5 minutes, before addition of 1 μM FisB ECD. Absorbance increases with increasing liposome aggregation due to increased scattering [106].

## Filamentous *B. subtilis* cells to test for curvature-sensitive localization of FisB

An overnight culture of BMB014 was diluted into fresh CH medium [107] to $OD_{600} = 0.05$. Moreover, 1 mM IPTG and 20 mM xylose were added to induce the expression of GFP-FisB and MciZ, respectively. The latter inhibits cytokinesis [70]. The culture was grown at 37˚C for 30 minutes before 3 to 5 μl of cells were transferred onto a 3% agar pad also containing 1 mM IPTG and 20 mM xylose. Cells were grown on the agar pad for 2 hours at 37˚C prior to imaging. GFP-FisB foci were detected using the ImageJ plugin TrackMate as described above. Radii of the inner and outer edges were determined by manually fitting a circle to the cells using ImageJ.

## Determination of binding constants

For determination of binding constants, the floatation protocol was slightly modified. Varying amounts of lipids were incubated with 100 nM iFluor555-FisB ECD for 1 hour at room temperature in a total volume of 100 μl. Density gradients were created as before using Optiprep (iodixanol); however, only 2 fractions were collected (Fig 5H). The protein concentration in fraction A was too small to be quantified by SDS-PAGE. Therefore, the sample was concentrated by trichloroacetic acid (TCA) precipitation. Briefly, 50 μl of TCA was added to fraction A and incubated for 30 minutes at 4˚C. The sample was spun at 14 krpm in an Eppendorf microfuge for 5 minutes. The pellet was washed twice with ice-cold acetone and subsequently dried for 10 minutes in a 95˚C heating block. A total of 10 μl of 2X SDS sample buffer was added to the dried pellet and the sample was boiled for 10 minutes at 95˚C and loaded on a 12% Bis-Tris gel. The amount of recovered protein was determined by fluorescence intensity of the labeled FisB ECD band on the gel using a Typhoon FLA 9500 (GE Healthcare). The dissociation constant $K_d$ was determined following [107]. Titration curves were fitted to

$$f_b = \frac{K[L]}{1 + K[L]},$$

where $f_b$ is the fraction of bound protein and $K$ the apparent association constant ($K = 1/K_d$). The equation above assumes that the total lipid concentration $[L]$ is much larger than the concentration of bound protein, a condition satisfied in our experiments for $[L] > 10^{-7}$ M.

## Image analysis

To estimate the fraction of cells that have undergone membrane fission at a particular time after sporulation was initiated by nutrient downshift, cells were labeled with TMA-DPH (see "Live-cell fluorescence microscopy of *B. subtilis*" above). The dye labels the forespore contours intensely before membrane fission, as it has access to 3 membranes in close proximity that cannot be resolved (forespore, engulfment, and mother cell membranes). After membrane fission, the dye dimly labels forespore contours (see S1 Appendix Fig A for examples and quantification). Due to the clear separation between the 2 labeling patterns (S1 Appendix Fig A), cells can be scored visually, with 6% to 7% of cells having intermediate labeling that prevents categorization. Thus, we underestimate the percentage of cells that have undergone membrane fission by at most approximately 7%.

For the analyses shown in S1 Appendix Figs D, panels A, B, C, E, and 9, we calculated the total intensity (sum ox pixel values) inside the cell contour (indicated in yellow in S4A Fig) using MicrobeJ [108]. Mean integrated autofluorescence (approximately 1,300 a.u) was

calculated by analyzing in the same way an equivalent number of individual wild-type cells, imaged under identical conditions.

For the analyses shown in Fig 2 and S1 Appendix Fig D, panel D, FisB foci were semiautomatically selected using SpeckleTrackerJ [48]. For each spot, the sum of pixel values in a $6 \times 6$ pixel (0.5 μm × 0.5 μm) box around the center of the spot were calculated. For each corresponding cell, the same operation was performed at a membrane area where no clusters were present and subtracted from the FisB cluster intensity.

### Preparation of DNA origami-based mEGFP standards

These standards were prepared and characterized as described in [44]. Briefly, DNA "rods" consisted of 6-helix bundle DNA origami nanotubes. Rods carried varying numbers of single stranded "handle" sequences for DNA-conjugated fluorophore hybridization. A long scaffold DNA (p7308 [109]) was folded into the desired shape by self-assembly with a 6-fold molar excess of designed "staple strands" by heating and cooling cycles over an 18-hour period in a thermocycler (Bio-Rad). Excess staples were removed by PEG precipitation [110], and DNA-conjugated fluorophores were hybridized to the DNA origami nanotubes by coincubation for 2 hours at 37°C. Finally, excess fluorophore-DNA conjugates were removed by a second PEG precipitation [110]. To estimate fluorophore labeling efficiency, standards designed to host 5 copies of Alexa Fluor 488 were similarly prepared. These standards were imaged on a TIRF microscope (Eclipse Ti, Nikon, Melville, NY, USA) until fully bleached. The photobleaching steps of the fluorescence traces were fit to a binomial function to estimate the labeling efficiency to be approximately 80% (95% CI = 76% to 84%).

### Quantitative western blot

mYFP was cloned into pVS001 (His$_6$-Sumo-mYFP) and purified using affinity chromatography. For immunoblotting, cells in 100-ml sporulation medium were pelleted and the supernatant removed. The pellets were suspended in ice-cold lysis buffer (pH = 7.5; 50 mM HEPES, 100 mM KCl, 3 mM MgCl$_2$, 1 mM EGTA, 0.1% Triton X-100, 1 mM DTT, 1 mM PMSF), with 1 complete protease inhibitor tablet (Roche, Basel, Switzerland)) to a final volume of 300 μl, and then we added 0.3 g acid-washed glass beads (425 to 600 μm, Sigma). After adding 150-μl boiling sample buffer (250 mM Tris-HCl, pH 6.8, 50% glycerol, 3.58 μM β-mercaptoethanol, 15% SDS, and 0.025% Bromophenol Blue), samples were incubated at 100°C for 5 minutes. Samples were centrifuged at 14,000 rpm in a desktop centrifuge at room temperature for 10 minutes and stored at −80°C. The blots were probed with peroxidase-conjugated anti-GFP antibody (ab13970). Images were scanned and quantified using ImageJ.

The numerical data used in all figures are included in S1 Data.

### Supporting information

**S1 Appendix. File containing Supporting experimental information, including quantification of FisB copy numbers using *B. subtilis* fluorescence standards and Western blotting, information showing that the localization of FisB is not coupled to cell wall remodeling, the proton motive force, or the membrane potential, data testing the topology of FisB, Figs A–L, and Tables A–C.** FisB, fission protein B.
(DOCX)

**S2 Appendix. Theoretical modeling, including Figs A–C.**
(PDF)

**S1 Data.** Excel spreadsheet containing, in separate sheets, the underlying numerical data and statistical analysis for Figs 1F, 1G, 1H, 2C, 2D, 2E, 3D, 3G, 3H, 4E, 4F, 4G, 5B, 5E, 5I, 5J, 6B, 6C, 6D, 7A, 7C, 7D, 7E, 7F, 8B, 8C, 8D, and 8E and Supporting information Figs 1C, 1E, 1G, 2B, 2C, 2D, 2E, 2F, 3A, 4B, 4D, 4E, 5, 6B, 6D, 6E, 6F, 8B, 9C, 9F, 9G, 9H, 10B, 11A, 11C, and 11D.
(XLSX)

**S1 Movie. FisB forms dynamic foci in the mother cell membranes.** mGFP-FisB forms dynamic foci in the mother cell membranes. The movie is a wide-field microscopy time-lapse movie of strain BAM003, cropped to show a single cell (the parent stack consisted of 1024 × 1024 pixels) at t = 2.5 hours after nutrient downshift. For determining the MSD of DMCs, we analyzed 60 different mobile FisB spots from 45 different bacteria. The focal plane was at the bottom surface of the cell. Images were acquired every 1 second. Scale bar 1 μm. Related to data in Fig 2E. DMC, dim mobile cluster; FisB, fission protein B; MSD, mean-squared displacement.
(AVI)

**S2 Movie. GFP-FisB foci are immobile at the cell pole at the time of membrane fission.** The movie is a wide-field microscopy time-lapse movie of strain BAM003, cropped to show a single cell (the parent stack consisted of 1024 × 1024 pixels) at t = 3 hours after nutrient downshift. For determining the MSD of ISEPs, we analyzed 30 different FisB spots at the tip from 30 different bacteria at T3. The focal plane was at mid-cell. Images were acquired every second. Scale bar 1 μm. Related to data in Fig 2E. FisB, fission protein B; ISEP, intense spot at the engulfment pole; MSD, mean-squared displacement.
(AVI)

**S3 Movie. B. subtilis cells expressing mGFP-FisB (strain BM0B14), imaged on a 2% PBS agarose pad.** Related to data in S1 Appendix Fig F, panels D–F. TIRF microscopy images were acquired at 1 frame/s, with excitation at 488 nm and exposure time 100 ms, using an Olympus IX81 inverted microscope equipped with an Andor ixon-ultra-897 EM CCD camera. Scale bar 5 μm. FisB, fission protein B.
(AVI)

**S4 Movie. A single cell from S3 Movie expressing mGFP-FisB.** Related to data in S1 Appendix Fig F, panel C, top row. Scale bar 5 μm. FisB, fission protein B.
(AVI)

**S5 Movie. B. subtilis cells expressing GFP-Mbl (strain BDR2061), imaged on a 2% PBS agarose pad.** Related to data in S1 Appendix Fig F, panels B, E, and F. TIRF microscopy images were acquired at 1 frame/s, with excitation at 488 nm and exposure time 50 ms, using an Olympus IX81 inverted microscope equipped with an Andor ixon-ultra-897 EM CCD camera. Scale bar 5 μm.
(AVI)

**S6 Movie. A single cell from S5 Movie expressing GFP-Mbl.** Related to data in S1 Appendix Fig F, panel A, top row. Scale bar 5 μm.
(AVI)

**S7 Movie. B. subtilis cells in the presence of 50 μg/ml fosfomycin expressing GFP-Mbl (strain BDR2061), imaged on a 2% PBS agarose pad.** Related to data in S1 Appendix Fig F, panels B, E, and F. TIRF microscopy images were acquired at 1 frame/s, with excitation at 488 nm and exposure time 50 ms, using an Olympus IX81 inverted microscope equipped with an

Andor ixon-ultra-897 EM CCD camera. Scale bar 5 μm.
(AVI)

**S8 Movie. A single cell from S7 Movie in the presence of 50 μg/ml fosfomycin expressing GFP-Mbl.** Related to data in S1 Appendix Fig F, panel A, bottom row. Scale bar 5 μm.
(AVI)

**S9 Movie. B. subtilis cells in the presence of 50 μg/ml fosfomycin expressing mGFP-FisB (strain BMB014), imaged on a 2% PBS agarose pad.** Related to data in S1 Appendix Fig F, panels D–F. TIRF microscopy images were acquired at 1 frame/s, with excitation at 488 nm and exposure time 100 ms, using an Olympus IX81 inverted microscope equipped with an Andor ixon-ultra-897 EM CCD camera. Scale bar 5 μm. FisB, fission protein B.
(AVI)

**S10 Movie. A single cell from S9 Movie in the presence of 50 μg/ml fosfomycin expressing mGFP-FisB.** Related to data in S1 Appendix Fig F, panel C, bottom row. Scale bar 5 μm. FisB, fission protein B.
(AVI)

## Acknowledgments

We thank members of the Karatekin and Rudner laboratories for stimulating discussions; Vladimir Polejaev and Jeorg Nikolaus (directors of the Yale West Campus Imaging Core) and Josh Lees (Yale Center for Cellular and Molecular Imaging Electron Microscopy Facility) for their help with imaging; Karin Reinisch in whose laboratory work by FH was carried out; Daniel R. Zeigler (Bacillus Genetic Stock Center) for helpful advice; and Alexander J. Meeske for some of the strains used in this study. We are grateful to Aurelien Roux (U. Genève) for the generous gift of Atto395-EndoA1.

## Disclaimers

The content is solely the responsibility of the authors and does not necessarily represent the official views of the National Institutes of Health.

## Author Contributions

**Conceptualization:** Ane Landajuela, Martha Braun, Alejandro Martínez-Calvo, Ned S. Wingreen, David Z. Rudner, Erdem Karatekin.

**Data curation:** Ane Landajuela, Martha Braun, Florian Horenkamp, Anna Andronicos, Vladimir Shteyn, Nathan D. Williams.

**Formal analysis:** Ane Landajuela, Martha Braun, Alejandro Martínez-Calvo, Vladimir Shteyn, Nathan D. Williams, Ned S. Wingreen, Erdem Karatekin.

**Funding acquisition:** Chenxiang Lin, Ned S. Wingreen, David Z. Rudner, Erdem Karatekin.

**Investigation:** Ane Landajuela, Martha Braun, Alejandro Martínez-Calvo, Florian Horenkamp, Anna Andronicos, Vladimir Shteyn, Nathan D. Williams, Ned S. Wingreen, Erdem Karatekin.

**Methodology:** Martha Braun, Christopher D. A. Rodrigues, Vladimir Shteyn, Nathan D. Williams, Ned S. Wingreen.

**Project administration:** Erdem Karatekin.

**Resources:** Christopher D. A. Rodrigues, Thierry Doan, Nathan D. Williams, Chenxiang Lin, David Z. Rudner, Erdem Karatekin.

**Software:** Erdem Karatekin.

**Supervision:** Chenxiang Lin, Ned S. Wingreen, David Z. Rudner, Erdem Karatekin.

**Validation:** Ane Landajuela, Martha Braun, Christopher D. A. Rodrigues, Alejandro Martínez-Calvo, Thierry Doan, Chenxiang Lin, Ned S. Wingreen, David Z. Rudner, Erdem Karatekin.

**Visualization:** Ane Landajuela, Martha Braun.

**Writing – original draft:** Erdem Karatekin.

**Writing – review & editing:** Ane Landajuela, Martha Braun, Christopher D. A. Rodrigues, Alejandro Martínez-Calvo, Thierry Doan, Florian Horenkamp, Anna Andronicos, Chenxiang Lin, Ned S. Wingreen, David Z. Rudner, Erdem Karatekin.

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
