## [Editor Report · Decision Letter 0]

23 Mar 2021

Dear Dr. Karatekin, 

Thank you for submitting your manuscript entitled "FisB relies on homo-oligomerization and lipid-binding to catalyze membrane fission in bacteria" for consideration as a Research Article by PLOS Biology. Sorry for the delay meanwhile we were considering your manuscript. 

Your manuscript has now been evaluated by the PLOS Biology editorial staff, as well as by an academic editor with relevant expertise, and I am writing to let you know that we would like to send your submission out for external peer review.

Please re-submit your manuscript within two working days, i.e. by Mar 25 2021 11:59PM.

Kind regards,

Paula

---

Associate Editor

PLOS Biology

---

## [Decision Letter · Decision Letter 1]

24 Apr 2021

Dear Dr. Karatekin,

Thank you very much for submitting your manuscript "FisB relies on homo-oligomerization and lipid-binding to catalyze membrane fission in bacteria" for consideration as a Research Article at PLOS Biology. Your manuscript has been evaluated by the PLOS Biology editors, an Academic Editor with relevant expertise, and by several independent reviewers.

In light of the reviews (below), we are pleased to offer you the opportunity to address the comments from the reviewers in a revised version that we anticipate should not take you very long. We will then assess your revised manuscript and your response to the reviewers' comments and we may consult the reviewers again.

You will see that the previous reviewers are satisfied with your revision. However, the modelling expert (reviewer #4), that is a new reviewer that we added to assess the newly incorporated modelling section, has some issues that need to be addressed. In particular, reviewer #4 thinks that the modelling section should be written with more clarity, asks why the bending energy is necessary and says that you need to give a better explanation of what energy is being minimized. This reviewer also asks about the fluctuations and the local hydrodynamics and whether you can comment on the role these factors may play, says that you fail to cite the relevant references in the fusion modelling field, says that you should mention the role that boundary conditions might play in altering the energy minimization, finds that explanations of the notations are missing and there is repetition in supplement and modelling sections, says you should comment on the limitations of the model, and has questions about the section on diffusion. Please address all issues raised by this reviewer.

Please also address the data and other policy-related requests that you will find in this letter below my signature, before the reviewers' comments.

We expect to receive your revised manuscript within 1 month.

**IMPORTANT - SUBMITTING YOUR REVISION**

*Resubmission Checklist*

*Published Peer Review*

*PLOS Data Policy*

*Blot and Gel Data Policy*

Sincerely,

Paula

---

Associate Editor,

pjaureguionieva@plos.org,

PLOS Biology

DATA POLICY:

Regardless of the method selected, please ensure that you provide the individual numerical values that underlie the summary data displayed in the following figure panels as they are essential for readers to assess your analysis and to reproduce it: Figures 1F, 1G, 1H, 2C, 2D, 2E, 3D, 3G, 3H, 4E, 4F, 4G, 5B, 5E, 5I, 5J, 6B, 6C, 6D, 7A, 7C, 7D, 7E, 7F, 8B, 8C, 8D, 8E, 9B, Supplementary figures 1C, 1E, 1G, 2B, 2C, 2D, 2E, 2F, 3A, 4B, 4D, 4E, 5, 6B, 6D, 6E, 6F, 8B, 9C, 9F, 9G, 9H, 10B, 11A, 11C, 11D.

**Please also ensure that figure legends in your manuscript include information on where the underlying data can be found**, and ensure your supplemental data file/s has a legend.

Please provide the original gel pictures for figures 3B, 4D, 5D, supplementary figures 3A, 3B, 4F, 8D, 9B, 9D, 10C.

Please add size bars for the microscopy pictures in supplementary figure 1D, 1F, and 11E, and indicate the size in the figure legend.

REVIEWS:

Reviewer #1: Differentiation and division mechanisms.

Reviewer #2: Rajesh Ramachandran. Molecular mechanisms of membrane remodeling, fission and fusion.

Reviewer #3: Patrick Eichenberger. Comparative and functional genomics of endospore-forming bacteria.

Reviewer #4: Modelling membrane curvature and membrane-protein interactions.

Reviewer #1: In the revised manuscript by Landajuela et al., the authors have adequately responded to all my original concerns with fresh experiments and a new quantitative simulation which proposes that FisB clustering and lipid binding can account for its subcellular localization. I have only one cosmetic suggestion for the authors, which they may or may not wish to incorporate.

Minor comment:

1. Fig. 8A. I assume that the tubulation by Endo A1, as seen by light microscopy, is the increase in fluorescence outside the immediate periphery of the liposome. Consider indicating this in the figure with an arrow (defined in the legend) so that a general reader may be alerted to the difference they are supposed to appreciate.

Reviewer #2: The authors have convincingly answered my queries with new Fig. 8 and 9. I have no further concerns and commend the investigators on an impressive piece of work.

Reviewer #3: The authors have done an excellent job addressing the suggestions of the reviewers.

Reviewer #4: I was asked to review a revised version of this manuscript and therefore, I focused only on the new section, which is the model. I leave it to the other three reviewers to comment on whether their queries were answered or not.

All my comments are restricted to the modeling section alone with the hope of helping the authors clarify many of the points they are making.

To address the concerns raised by the previous round of reviews, the authors adding a physical model to explain how the trans-aggregation of FisB can be energetically favorable. I have the following comments for the authors to consider. Overall, I think the modeling section should be written with more clarity to better serve its purpose in this paper. There are parts that are quite confusing preventing the reader from fully appreciating the value proposition that the model makes.

1) The free energy of the membrane is written as a bending energy plus the energy of membrane-protein interactions. What is interesting is that there are two separate membranes that are trying to fuse together to complete the fission process and this fusion of these two membranes is mediated by these trans interactions. I ask because from all the schematics (Fig 1A i to Fig 9 and the supplementary material figures), this is what is depicted. Since the membrane is assumed to be locally flat and uncoupled from the curvature, why is the bending energy necessary? I feel like I'm missing some explanation that should be clarified in what energy is being minimized (literally — what is the system and what are the surroundings?). I wonder if the issue can be resolved by writing the energies of membrane domain 1, membrane domain 2 and the membrane-protein adhesion energies for clarity but perhaps the authors have better ideas.

2) So if it is two separate membranes trying to fuse, then aren't fluctuations and the local hydrodynamics important? Can the authors comment on the role these factors may play?

3) Essentially, the authors are using a LJ potential and a repulsion energy U(\\phi) to capture the adhesion interactions of the trans-FisB molecules and minimize the cis interactions. This is fine but again, why not cite the relevant references that have actually considered this problem at various levels of complexities? I might add that this is true of the entire modeling section — it is rather sparse on references in the field, even missing the seminal ones that have agonized about fusion etc.

4) Boundary conditions — the authors should mention that role that boundary conditions might play in altering the energy minimization. No mention is made of the edge effects, which are going to be critical in determining the neck.

5) notation — what are \\sigma and a? I couldn't find any explanation for all the notation used. Also, much of the supplement and the modeling section are near verbatim in many paragraphs. Please edit to remove repetition.

6) One other curiosity is that the authors predict a radius of 2.7 nm, which is roughly half the thickness of the bilayer and well below the applicability of the Helfrich model. Can the authors comment on the limitations of their model?

7) section on diffusion — shouldn't the Laplacian operator be a Laplace-Beltrami operator on the surface? Also, in the absence of large curvatures and curvature-mediated interactions, why would diffusion be considered rate limiting? This is also another section that is missing many references.

---

## [Editor Report · Decision Letter 2]

7 Jun 2021

Dear Dr. Karatekin,

On behalf of my colleagues and the Academic Editor, Frederick Hughson, I am pleased to say that we can in principle offer to publish your Research Article "FisB relies on homo-oligomerization and lipid-binding to catalyze membrane fission in bacteria" in PLOS Biology, provided you address any remaining formatting and reporting issues. These will be detailed in an email that will follow this letter and that you will usually receive within 2-3 business days, during which time no action is required from you. Please note that we will not be able to formally accept your manuscript and schedule it for publication until you have made the required changes.

PRESS

Thank you again for supporting Open Access publishing. We look forward to publishing your paper in PLOS Biology. 

Sincerely, 

Paula

---

Paula Jauregui, PhD 

Associate Editor 

PLOS Biology